# Oncogenic drivers dictate immune control of acute myeloid leukemia

Rebecca J. Austin[1,2,3,4,11], Jasmin Straube[1,2,11], Rohit Halder[1],
Yashaswini Janardhanan[1], Claudia Bruedigam[1,2], Matthew Witkowski[3,4,5],
Leanne Cooper[1], Amy Porter[1], Matthias Braun[1],
Fernando Souza-Fonseca-Guimaraes[6], Simone A. Minnie[1,7], Emily Cooper[1],
Sebastien Jacquelin[1,8], Axia Song[1], Tobias Bald[1,9], Kyohei Nakamura[1],
Geoffrey R. Hill[1,7], Iannis Aifantis[3,4], Steven W. Lane[1,2,10,12] ✉ &
Megan J. Bywater[1,2,12] ✉

Acute myeloid leukemia (AML) is a genetically heterogeneous, aggressive hematological malignancy induced by distinct oncogenic driver mutations. The effect of specific AML oncogenes on immune activation or suppression is unclear. Here, we examine immune responses in genetically distinct models of AML and demonstrate that specific AML oncogenes dictate immunogenicity, the quality of immune response and immune escape through immunoediting. Specifically, expression of Nras[G12D] alone is sufficient to drive a potent anti-leukemia response through increased MHC Class II expression that can be overcome with increased expression of Myc. These data have important implications for the design and implementation of personalized immu-notherapies for patients with AML.

Acute myeloid leukemia (AML) is caused by the acquisition of genetic mutations in hematopoietic stem and progenitor cells (HSPCs) resulting in a block in myeloid differentiation and the expansion of immature myeloid blasts[1]. AML is genetically heterogeneous with recurrent genetic abnormalities resulting in activation of signal transduction pathways, impaired function of lineage-specific transcription factors and dysregulation of epigenetic modifiers[2]. Survival and response to chemotherapy is dependent on the age and molecular profile of AML patients[3,4]. Despite chemotherapy, followed where possible by allo-geneic hematopoietic stem cell transplantation (allo-HSCT), or low-intensity combination therapies for elderly patients[5], long-term survival is <50% overall and is attributed to relapse or therapy resistance highlighting the importance of developing novel therapies.

There is increasing evidence to support a functional interaction between AML and the immune system[6,7]. AML patients exhibit myeloid dysfunction, cytotoxic lymphocyte dysfunction of both NK and T cells, secretion of suppressive molecules and upregulation of immune suppressive ligands on AML cells[8–13]. Studies indicate that immune microenvironment composition is also important for response to chemotherapeutic treatments. For example, AML patients with abnormal NK cell function and downregulation of NK cytotoxicity surface receptors have defective NK clearance of leu-kemic blasts[14–17]. Lymphocyte recovery after chemotherapy is associated with improved survival and there are even rare cases of spontaneous remission after severe infections[18,19]. Early clinical trials suggest that combining hypomethylating agents and immune

[1]Cancer Program, QIMR Berghofer Medical Research Institute, Brisbane 4006, Australia. [2]The University of Queensland, St Lucia, Brisbane, QLD 4072, Australia. [3]Department of Pathology, NYU Grossman School of Medicine, New York, NY 10016, USA. [4]Laura & Isaac Perlmutter Cancer Center, NYU Grossman School of Medicine, New York, NY 10016, USA. [5]Department of Pediatrics, University of Colorado Anschutz Medical Campus, Aurora, CO 80045, USA. [6]Frazer Institute, The University of Queensland, Woolloongabba, QLD 4102, Australia. [7]Translational Science and Therapeutics Division, Fred Hutchinson Cancer Centre, Seattle Cancer Care Alliance, Seattle, WA, USA. [8]Mater Research, Translational Research Institute, The University of Queensland, Woolloongabba, QLD 4102, Australia. [9]Institute of Experimental Oncology, University Hospital of Bonn, 53127 Bonn, Germany. [10]Cancer Care Services, Royal Brisbane and Women's Hospital, Herston 4029, Australia. [11]These authors contributed equally: Rebecca J. Austin, Jasmin Straube. [12]These authors jointly supervised this work: Steven W. Lane, Megan J. Bywater. ✉e-mail: steven.lane@qimrberghofer.edu.au; megan.bywater@qimrberghofer.edu.au

checkpoint inhibitors may have efficacy in AML, however these results have not yet been confirmed in randomized studies[20,21]. AML has a low somatic mutation burden and is predicted to have a low frequency of potential neoantigens[22,23]. This poses the question, what regulates immune responses in AML and can distinct genetic aberrations influence immunogenicity? Characterization of the immune microenvironment in specific types of AML, including the mechanisms of immune escape, may help to understand whether the endogenous immune response is capable of controlling, or eliminating AML.

Here, we investigate the immunogenicity of genetically distinct models of AML, representing common clinical and prognostic subsets of genetic alterations found in AML patients[24,25]. We find that distinct oncogenes alter the host immune response to the leukemia and that mutant Nras is a key determinant of this immunological selection[26,27].

Altogether, these data provide insights into endogenous anti-leukemia immune responses in AML and generate a path for the strategic use of immunotherapies for subsets of AML patients.

## Results

### Oncogene specificity defines AML immune response

We generated three genetically distinct models of AML; BCR-ABL + NUP98-HOXA9 (BA/NH), MLL-AF9 (MA9) or AML1-ETO + Nras$^{G12D}$ (AE/Nras$^{G12D}$), representative of common genetic alterations found in patients with AML[3] (Fig. 1A). Rag2$^{-/-}$γc$^{-/-}$ donors were used to avoid the transfer of mature lymphoid immune cells from the donor graft[28]. Rag2$^{-/-}$γc$^{-/-}$ and wild-type C57BL/6J (WT) had similar HSPC baseline function assessed by colony formation in cytokine-enriched methyl-cellulose (Supplementary Fig. 1A)[1]. Primary (1°) AML developed in all recipients reconstituted with AML oncogene-expressing BM (Supplementary Fig. 1B) but not with HSPCs transduced by retrovirus without

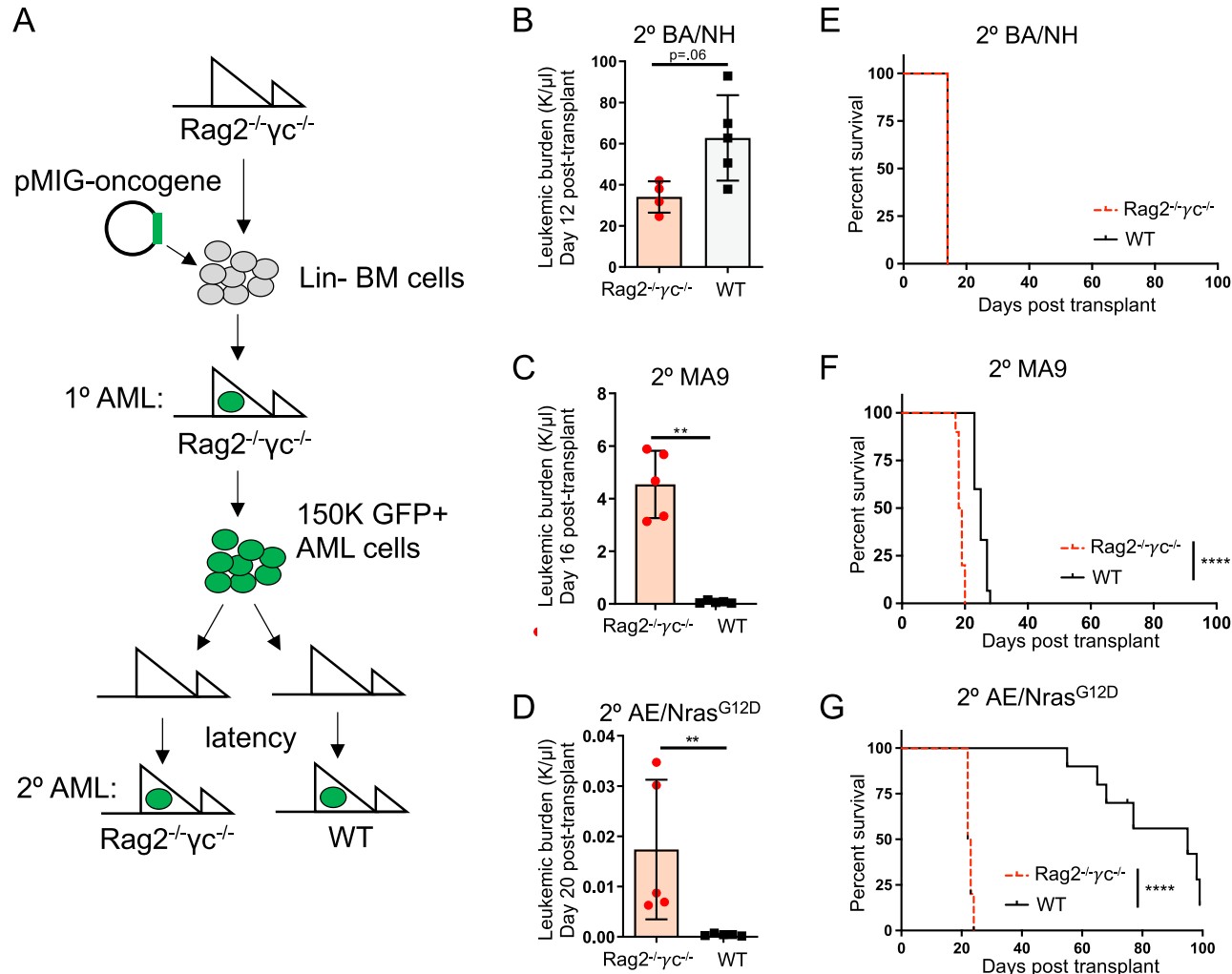

**Fig. 1 | Oncogene specificity dictates AML immunogenicity. A** Schema for in vivo generation of oncogene-specific retroviral 1° AMLs in Rag2$^{-/-}$γc$^{-/-}$ mice, then passaged through 2° Rag2$^{-/-}$γc$^{-/-}$ or wildtype (WT) C57BL/6J mice. Peripheral blood (PB) leukemic burden (WBC × GFP%) of secondary recipients: **B** BA/NH day 12 post-transplant (n = 4 (Rag2$^{-/-}$γc$^{-/-}$), n = 5 (wildtype), p = 0.0635), **C** MA9 day 16 post-transplant (n = 5, p = 0.0079), **D** AE/Nras$^{G12D}$ on day 20 post-transplant (n = 5, p = 0.0079). Data are presented as mean values +/− SD, from one of two repeat experiments. Kaplan–Meier curves comparing survival between Rag2$^{-/-}$γc$^{-/-}$ and WT secondary recipients transplanted with 150,000 1° AML cells demonstrating, **E** BA/NH (Rag2$^{-/-}$γc$^{-/-}$ n = 4; WT n = 5; median survival 14 days), **F** MA9 (Rag2$^{-/-}$γc$^{-/-}$ n = 10; WT n = 10; median survival 18 vs. 25 days respectively, p < 0.0001) and, **G** AE/Nras$^{G12D}$ (Rag2$^{-/-}$γc$^{-/-}$ n = 10; WT n = 10; median survival 22 vs. 95 days respectively, p < 0.0001). BA/NH demonstrating data from one experiment. MA9 and AE/Nras$^{G12D}$ demonstrating pooled data from two experiments. Each point represents a biologically independent animal. Two-sided Mann–Whitney test for comparison between two groups (**B**–**D**) and Mantel-Cox test for comparison of Kaplan–Meier curves (**E**–**G**). *p < 0.05, **p < 0.01, ***p < 0.001, ****p < 0.0001. Source data are provided as a Source Data file.

oncogene expression. AML generated from the dual transduction of oncogenes were genotyped to confirm the integration of both vectors (Supplementary Fig. 1C).

In order to determine the effect of the immune system on disease progression, 1° AMLs were transplanted into secondary (2°) non-irradiated immunodeficient Rag2$^{-/-}$γc$^{-/-}$ or immunocompetent WT recipients (Fig. 1B–G). BA/NH induced AML in either WT or Rag2$^{-/-}$γc$^{-/-}$ recipients with similar overall survival (Fig. 1B, E, Supplementary Fig. 1D). In contrast, MA9 AML and AE/Nras$^{G12D}$ AML progressed more rapidly in Rag2$^{-/-}$γc$^{-/-}$ compared to WT recipients (Fig. 1C, D, F–G, Supplementary Fig. 1E, F) with AE/Nras$^{G12D}$ AML showing the most prolonged latency in immunocompetent hosts. BM AML engraftment was similar between Rag2$^{-/-}$γc$^{-/-}$ and WT at 24hrs post-transplant, indicating similar homing to BM (Supplementary Fig. 1G). Furthermore, extending disease latency in BA/NH AML with the transplantation of fewer cells did not increase disease latency in WT recipients in comparison to Rag2$^{-/-}$γc$^{-/-}$ (Supplementary Fig. 1H). These data indicate a graded immune response to AML subtypes that is specified by individual oncogenes.

## Oncogene specificity influences mediators of immune recognition and immune activity

We next determined if the differences in AML immunogenicity are reflected through cell intrinsic differences in the expression of cell surface immune recognition markers by comparing the immunophenotype of the genetically distinct AMLs maintained exclusively in immunodeficient Rag2$^{-/-}$γc$^{-/-}$ recipients. Analysis restricted to the AML CD11b$^+$ myeloid population (Supplementary Fig. 2A) showed that AE/Nras$^{G12D}$ was characterized by the highest expression of antigen presentation machinery, H2-D$^b$, H2-K$^b$ and MHC Class II (Fig. 2A, Supplementary Fig. 2B). Unexpectedly, after serial transplantation the AE/Nras$^{G12D}$ AML passaged in both the Rag2$^{-/-}$γc$^{-/-}$ and WT 2° recipients only retained integration of the Nras$^{G12D}$ construct (hereafter referred to as Nras$^{G12D}$) (Supplementary Fig. 2C). Consistent with this, analysis of Rag2$^{-/-}$γc$^{-/-}$ HSPCs at 72h post transduction with the individual oncogenes, demonstrated that acute expression of Nras$^{G12D}$ alone was sufficient to drive increased surface expression of MHC Class II (Supplementary Fig. 2D). These findings were further confirmed using microarray data of bulk AML samples at diagnosis from 42 patients with an NRAS mutation compared to 10 patients with an MLL translocation (MLL-X). Here, NRAS mutant human AML showed increased expression of multiple HLA (MHC Class II) genes, including HLA-DQA1, compared to MLL-X translocated AML (Fig. 2B, Supplementary Fig. 2E). Using single sample gene set enrichment, MHC Class II score was associated with AML patients driver mutations and chromosomal aberrations[29], with NRAS mutant patients ranked as one of the highest MHC Class II expressing genetic groups while MLL-X translocated patients rank among the lowest (Fig. 2C). Single cell RNA-sequencing[30] demonstrated that malignant CD33-expressing AML blasts from RAS mutant patients maintained MHC Class II gene expression, whereas malignant CD33-expressing blasts from MLL-X AML patient samples had reduced expression in comparison to healthy CD33-expressing BM (Fig. 2D).

Next, we examined the difference in the expression of discrete panel of immunomodulatory cell surface molecules in murine AML samples. We found that Nras$^{G12D}$ AML was characterized by the highest expression of the CD28 ligands, CD80 and CD86 (Fig. 2E, Supplementary Fig. 3A). Conversely, BA/NH had high expression of the immunosuppressive ligands PD-L1 and GAL-9 and CD155 (Fig. 2E, Supplementary Fig. 3A). Analysis of Rag2$^{-/-}$γc$^{-/-}$ HPSCs at 72 h post transduction with the individual oncogenes, demonstrated that acute expression of Nras$^{G12D}$ alone is sufficient to drive increased surface expression of CD80 and CD86 (Supplementary Fig. 3B). Conversely, neither acute expression of BCR-ABL nor

NUP98-HOXA9 alone was sufficient to increase PD-L1, GAL-9 or CD155 (Supplementary Fig. 3B).

We next sought to determine if evidence of an anti-AML immune response was present in a genetically engineered knockin model of mutant Nras-driven AML, derived from mice heterozygous for conditional alleles conferring a C-terminal truncation in Npm1 and constitutively active Nras (Tg(MxI-cre), Npm1$^{fl-cA/+}$; Nras$^{fl-G12D/+}$), expressed from their respective endogenous promoters[31]. We expanded the AML by transplantation into immunodeficient mice and then transplanted this AML into tertiary (3°) non-irradiated immunodeficient Rag2$^{-/-}$γc$^{-/-}$ or immunocompetent WT recipients. Npm1c/Nras$^{G12D}$ AML cells generated a rapid, fully penetrant AML when transplanted into Rag2$^{-/-}$γc$^{-/-}$ recipients, however there was a marked delay in disease latency in immunocompetent WT recipients (Fig. 2F, G), confirming an intrinsic immune response to a genetically engineered AML mouse model driven by mutant Nras.

These data reveal discrete effects of oncogenic drivers on immune regulatory molecule expression in AML cells, supporting a model whereby Nras$^{G12D}$ AML has greater potential to interact with the immune system.

## Oncogene specificity determines the composition of the AML immune microenvironment

Given the role of MHC, CD80 and CD86 in T cell activation, we compared the requirement for T cells in controlling disease progression in MA9 and Nras$^{G12D}$ AML. WT recipient mice were treated with isotype control, or antibodies that depleted T cells (CD4$^+$ and CD8$^+$). Immune cell depletion was verified in the peripheral blood (Supplementary Fig. 4A). In Nras$^{G12D}$, depletion of T cells accelerated the development of AML (Fig. 3A). Similar findings were observed in the MA9 model but this effect was much less pronounced (Fig. 3B). Consistent with a specific T cell mediated immune response, we demonstrated that Nras$^{G12D}$ AML (Rag2$^{-/-}$γc$^{-/-}$) increases T cell proliferation upon co-culture in comparison to non-transformed BM (Rag2$^{-/-}$γc$^{-/-}$) whereas MA9 AML did not (Fig. 3C, D).

We next sought to determine if this differential requirement for T cells in the anti-leukemic response was reflected in the composition of the immune microenvironment. We observed a significant decrease in the frequency of T cells within the microenvironment of the leukemia-bearing spleens of BA/NH recipients compared to MA9 and Nras$^{G12D}$ recipients and naïve controls (Fig. 3E). We note that the BA/NH recipients demonstrated complete effacement of splenic architecture concordant with this loss of normal T-cell populations. Within T cells, Nras$^{G12D}$ recipients display the lowest percentage of CD4$^+$ T cells and subsequently the highest frequency of CD8$^+$ T cells (Fig. 3E). Interestingly, all AML recipients display contraction in the proportion of naïve CD4$^+$ and CD8$^+$ cells (CD44$^-$CD62L$^+$), an expansion in CD4$^+$ and CD8$^+$ T effector memory (CD44$^+$CD62L$^-$) and a decrease in CD4$^+$ and CD8$^+$ T central memory (CD44$^+$CD62L$^+$) formation when compared to naïve controls (Fig. 3E, Supplementary Fig. 4B–D). Of note however, is that BA/NH recipients retain the greatest frequency of naïve CD8$^+$ T cells, with Nras$^{G12D}$ recipients having a greater frequency of CD8$^+$ T effector memory compared to BA/NH recipients (Fig. 3E, Supplementary Fig. 4D).

As these immunocompetent recipients were analyzed after developing overt AML, we compared the impact of AMLs driven by distinct oncogenic drivers on markers of CD4$^+$ and CD8$^+$ T cell activation and dysfunction. There was an increase in the frequency of PD-1$^+$/DNAM-1$^+$ CD4$^+$ and CD8$^+$ T cells in AML recipients compared to naïve spleen, indicating an expansion of effector T cells with reduced cytotoxic potential (Fig. 3F, Supplementary Fig. 5A–D). However, the co-expression of co-inhibitory receptors KLRG1 and PD-1 was significantly increased on CD4$^+$ and CD8$^+$ T cells from Nras$^{G12D}$ recipients only compared to naïve spleen (Fig. 3F, Supplementary Fig. 5A–D). Finally, co-expression of PD-1

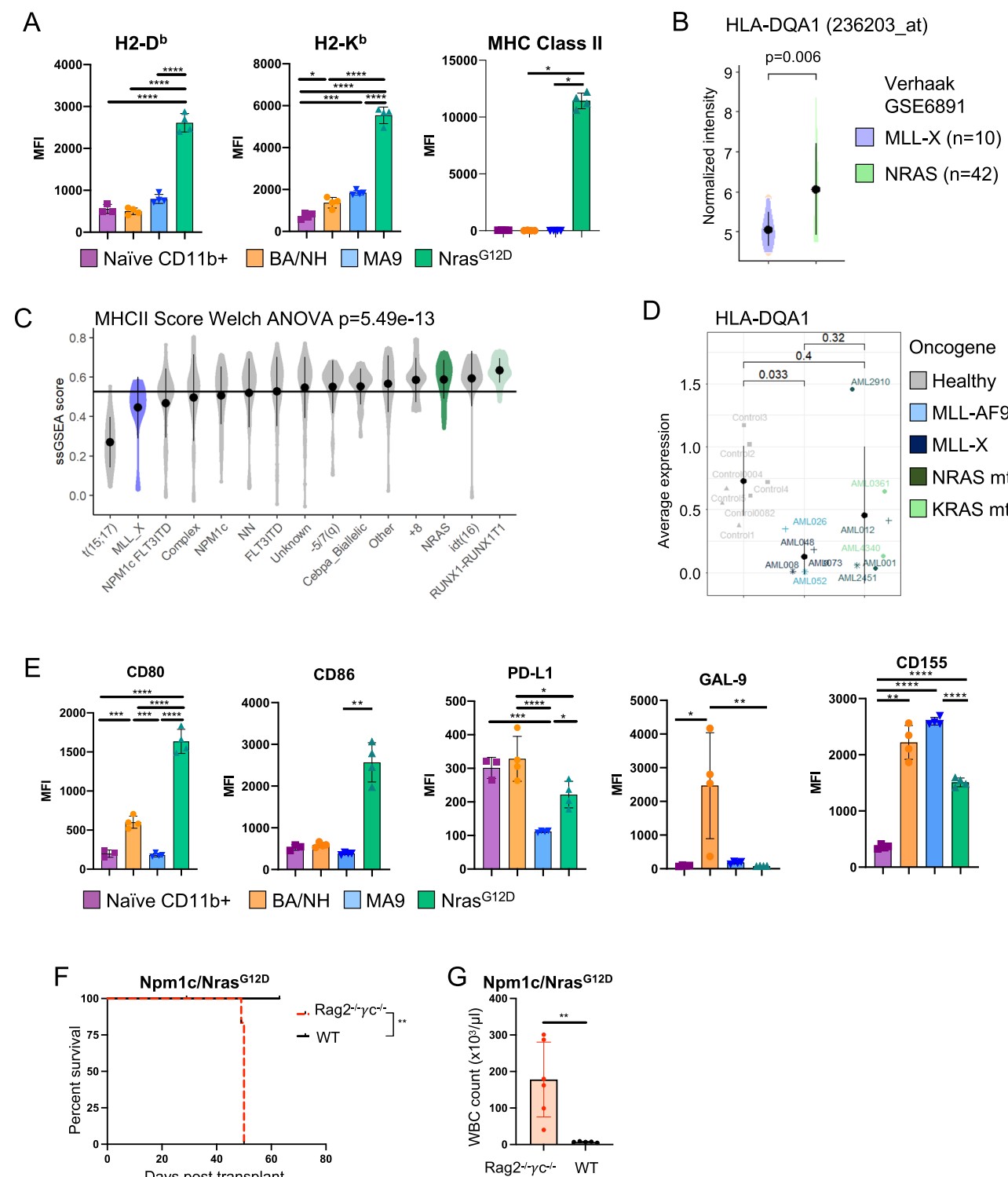

and TIM-3, indicating a T cell exhaustion phenotype, was increased on CD4+ and CD8+ T cells from NrasG12D recipients, but was only increased on the CD4+ T cells in BA/NH recipients and not increased on T cells from MA9 AML (Fig. 3F, Supplementary Fig. 5A–D). We sought to validate these murine findings in human AML using single cell RNA-sequencing analysis[30]. Consistent with the murine findings, we observed that greater frequency of T cells isolated from the bone marrow of mutant RAS AML patients demonstrated PD-1 gene expression in comparison to those isolated from MLL-X translocated AML patients (Fig. 3G). These data indicate expansion and dysfunction of the effector T cell

compartment as a distinguishing feature of the immune microenvironment of immunogenic AML.

### Immunoediting selects against immunogenic AML

Despite a robust immune response that delayed AML onset, MA9 and NrasG12D leukemias were eventually able to develop in the presence of a competent immune system. We hypothesized that this immune escape could be mediated through immunoediting, the selection of disease with decreased immunogenicity[26,27] or via immunosuppressive effects on the host immune system. To functionally examine for immunoediting, we compared the disease latency of AML passaged through

**Fig. 2 | Oncogene specificity influences the immunogenicity of AML cells.**
**A** Median fluorescence intensity (MFI) of H2-D$^b$ ($n=3$ (Naïve CD11b$^+$), $n=4$ (BANH, MA9, Nras$^{G12D}$)), H2-K$^b$ ($n=4$, $p=0.0167$ (Naïve vs BA/NH), $p=0.0002$ (Naïve vs MA9)), MHC Class II (IA/E) ($n=4$) on cell surface of myeloid cells (CD11b$^+$) from bone marrow (BM) of naïve wild-type mice and GFP$^+$ CD11b$^+$ BA/NH, MA9 and Nras$^{G12D}$ AML cells in the spleens of Rag2$^{-/-}$γc$^{-/-}$ recipients moribund with disease. Each point represents a biologically independent animal transplanted with the same tumor per genotype. Data are presented as mean values +/− SD. **B** Microarray derived gene expression of HLA-DQA1 from bulk PB/BM of 10 MLL-translocated (MLL-X) and 42 NRAS mutant AML patients[29]. Data are presented as mean values +/− SD.
**C** MHC Class II ssGSEA across Verhaak dataset genotypes. Data are presented as mean values +/− SD. **D** HLA-DQA1 gene expression in malignant CD33-expressing AML blasts derived from single cell RNA-sequencing of BM from patients with a MLL-X AML ($n=5$) or a mutant RAS AML ($n=6$) in comparison to CD33-expressing cells from healthy bone marrow ($n=7$)[30]. Data are presented as mean values +/− SD. **E** MFI of CD80 ($n=3$ (Naïve CD11b$^+$), $n=4$ (BA/NH, MA9, Nras$^{G12D}$), $p=0.0007$ (Naïve vs BA/NH), $p=0.0003$ (BA/NH vs MA9)), CD86 ($n=3$ (Naïve CD11b$^+$), $n=4$ (BA/NH, MA9, Nras$^{G12D}$), $p=0.0030$ (MA9 vs Nras$^{G12D}$)), PD-L1 ($n=3$ (Naïve CD11b$^+$), $n=4$ (BA/NH, MA9, Nras$^{G12D}$), $p=0.0006$ (Naïve vs MA9), $p=0.0212$ (BA/NH vs Nras$^{G12D}$), $p=0.0172$

(MA9 vs Nras$^{G12D}$)), GAL-9 ($n=4$, $p=0.0451$ (Naïve vs BA/NH), $p=0.0065$ (BA/NH vs Nras$^{G12D}$)) and CD155 ($n=4$, $p=0.0045$ (Naïve vs BA/NH)) on cell surface of myeloid cells (CD11b$^+$) from BM of naïve wild-type mice and GFP$^+$ CD11b$^+$ BA/NH, MA9 and Nras$^{G12D}$ AML cells in the spleens of Rag2$^{-/-}$γc$^{-/-}$ recipients moribund with disease. Each point represents a biologically independent animal transplanted with the same tumor per genotype. Data are presented as mean values +/− SD. **F** Kaplan−Meier curves comparing survival between Rag2$^{-/-}$γc$^{-/-}$ ($n=6$) and wildtype (WT, $n=6$) recipients transplanted with equal numbers of AML cells derived from a moribund *Tg(Mx1-cre); Npm1$^{fl-cA/+}$; NRas$^{fl-G12D/+}$ mouse* (Npm1c/Nras$^{G12D}$) ($p=0.0023$)[31]. **G** WBCC of mice transplanted with Npm1c/Nras$^{G12D}$ AML when moribund (Rag2$^{-/-}$γc$^{-/-}$, $n=6$) or 63 days post-transplant (WT, $n=5$) ($p=0.0094$). Each point represents a biologically independent animal transplanted with the same tumor per genotype. Data are presented as mean values +/− SD. One-way ANOVA (**C**) with Tukey's multiple testing correction (**A**: H2-D$^b$, H2-K$^b$, **E**: CD80, CD86), Mann−Whitney test for pairwise comparisons between groups (**B**), Kruskal−Wallis test with Dunn's multiple comparisons test (**A**: MHC Class II, **D**, **E**: CD86, GAL-9), Welch ANOVA with Dunnett's T3 multiple comparisons test (**E**: CD155), unpaired *t*-test with Welch's correction (**G**), Mantel-Cox test for comparison of Kaplan−Meier curves (**F**). *$p<0.05$, **$p<0.01$, ***$p<0.001$, ****$p<0.0001$. Source data are provided as a Source Data file.

immunocompetent mice vs. AML passaged through immunodeficient mice when these were transplanted into either an immunocompetent WT or immunodeficient Rag2$^{-/-}$γc$^{-/-}$ recipients (Fig. 4A). For both leukemias, there was no difference in disease latency when transplanted into Rag2$^{-/-}$γc$^{-/-}$ mice, suggesting that passage through an immunocompetent host does not change the proliferative capacity of these AMLs (Fig. 4B). However, there was accelerated disease progression in immunocompetent mice transplanted with Nras$^{G12D}$ AML that had been previously passaged through immunocompetent WT mice (2º WT; 3º WT), compared to AML passaged previously through immunodeficient Rag2$^{-/-}$γc$^{-/-}$ mice (2º Rag2$^{-/-}$γc$^{-/-}$; 3º WT) (Fig. 4B). In contrast, disease latency in immunocompetent mice was unchanged for MA9 AML, regardless of whether the AML was previously passaged through immunocompetent (2º WT; 3º WT) or immunodeficient (2º Rag2$^{-/-}$γc$^{-/-}$; 3º WT) mice, suggesting that immunoediting is not observed in MA9 AML (Fig. 4C) and reflecting the more immunogenic phenotype of Nras$^{G12D}$ (Fig. 1F, G). This demonstrates that Nras$^{G12D}$ AML is immunoedited during passage through immunocompetent recipients.

Given the elevated expression of a number of immune regulatory molecules of the surface of Nras$^{G12D}$ AML (Fig. 2A, D and Supplementary Figs. 2B, 3A), we used flow cytometry to examine the immunophenotype of non-immunoedited (N-IE) versus immunoedited (IE) Nras$^{G12D}$ cells (Supplementary Fig. 6A). Surprisingly, we didn't observe any difference in the abundance of H-2D$^b$ and MHC Class II, and only a minor increase in H-2K$^b$ (Fig. 4D). In contrast, we saw the upregulation of ligands with potential immunosuppressive function, PD-L1 and CD86 (Fig. 4E). These findings suggest that immunoedited Nras$^{G12D}$ AML may suppress the anti-leukemic immune response to facilitate disease progression.

As PD-L1 interacts with the immune-suppressive receptor PD-1 on T cells, we determined whether blockade of the PD-1/PD-L1 interaction with anti-PD-1 was able to reactivate an anti-leukemic immune response in immunoedited Nras$^{G12D}$ AML (Fig. 4F). We observed minor, but significant effects on AML control after anti-PD-1 antibody treatment. Anti-PD-1 treated mice showed a lower penetrance of AML infiltration in the liver, the major site of infiltration of this disease in unconditioned immunocompetent mice, with fewer tumors per liver and a trend to reduced tumor area (Fig. 4G). These data demonstrate that restoring immune cell function through anti-PD-1 treatment can facilitate anti-leukemic control, but has limited efficacy in established disease when used as a single agent. This is consistent with published data showing only a modest effect of PD-1 blockade in MDS and AML[20].

As modulating the surface immune checkpoints were insufficient to reinstate immune control in immunoedited Nras$^{G12D}$ AML, we used RNA-sequencing to investigate transcriptional changes between non-

immunoedited and immunoedited Nras$^{G12D}$ AML (Supplementary Fig. 6A, B, Supplementary Data 1). Surprisingly, the gene expression of immunoedited AMLs showed evidence of pathway down-regulation for NRAS signaling (Fig. 5A, Supplementary Data 1), correlating with decreased gene expression of Nras compared to non-immunoedited cells (Supplementary Fig. 6C, D, Supplementary Data 1). Q-RT-PCR analysis of genomic DNA of non-immunoedited and immunoedited Nras$^{G12D}$ cells revealed that immunoediting selected for cells with reduced Nras$^{G12D}$ copy number (Fig. 5B). Furthermore, the immunoedited AMLs were characterized by the increased expression of targets of MYC, a transcription factor conventionally considered to have a master regulatory role in growth and proliferation (Fig. 5C, Supplementary Data 1). Nras$^{G12D}$ AML in WT secondary recipients showed only a slightly higher frequency of Ki67 positive cells, with no difference in the mitotic marker phospho-Histone H3 (Supplementary Fig. 6E, F). Consistent with this finding, transplant of equal numbers of non-immunoedited and immunoedited Nras$^{G12D}$ cells into immunodeficient Rag2$^{-/-}$γc$^{-/-}$ mice generated disease with comparable latency suggesting that no intrinsic differences in proliferative capacity occurred as a consequence of immunoediting (Fig. 4B). Importantly, changes in NRAS and MYC gene sets were not observed when comparing MA9 AML passaged through WT vs Rag2$^{-/-}$γc$^{-/-}$ recipients (Supplementary Fig. 6G, Supplementary Data 1). However, we did observe that MA9 AMLs exposed to a competent immune environment upregulated the expression of genes involved in interferon signaling and that this did not occur in the immunoedited Nras$^{G12D}$ AML (Supplementary Fig. 6G−I, Supplementary Data 1). These data indicate that immunoediting in Nras$^{G12D}$ AML may involve coordinate clonal selection for cells with reduced mutant Nras expression and increased MYC-driven transcription to evade immune control.

## Ectopic expression of Myc reduces immunogenicity in Nras$^{G12D}$-driven AML

There is increasing evidence that MYC is a critical determinant of an immunosuppressive tumor microenvironment and that MYC inactivation enables recruitment of lymphocytes into tumors[32–35]. Consistent with this, in NRAS-mutant AML, we observed that a tissue-agnostic set of MYC transcriptional targets showed an inverse correlation with a predictive gene signature of cytotoxic immune cell infiltration in AML (Fig. 5D, Supplementary Data 2)[36,37].

To functionally examine the consequence of elevated Myc expression on immune evasion in AML, we generated Nras$^{G12D}$ AML expressing ectopic levels of Myc in comparison to an empty vector (EV) control. Dual oncogene integration was confirmed by expression of both GFP (Nras$^{G12D}$) and mCherry (Myc) fluorescent markers. Ectopic

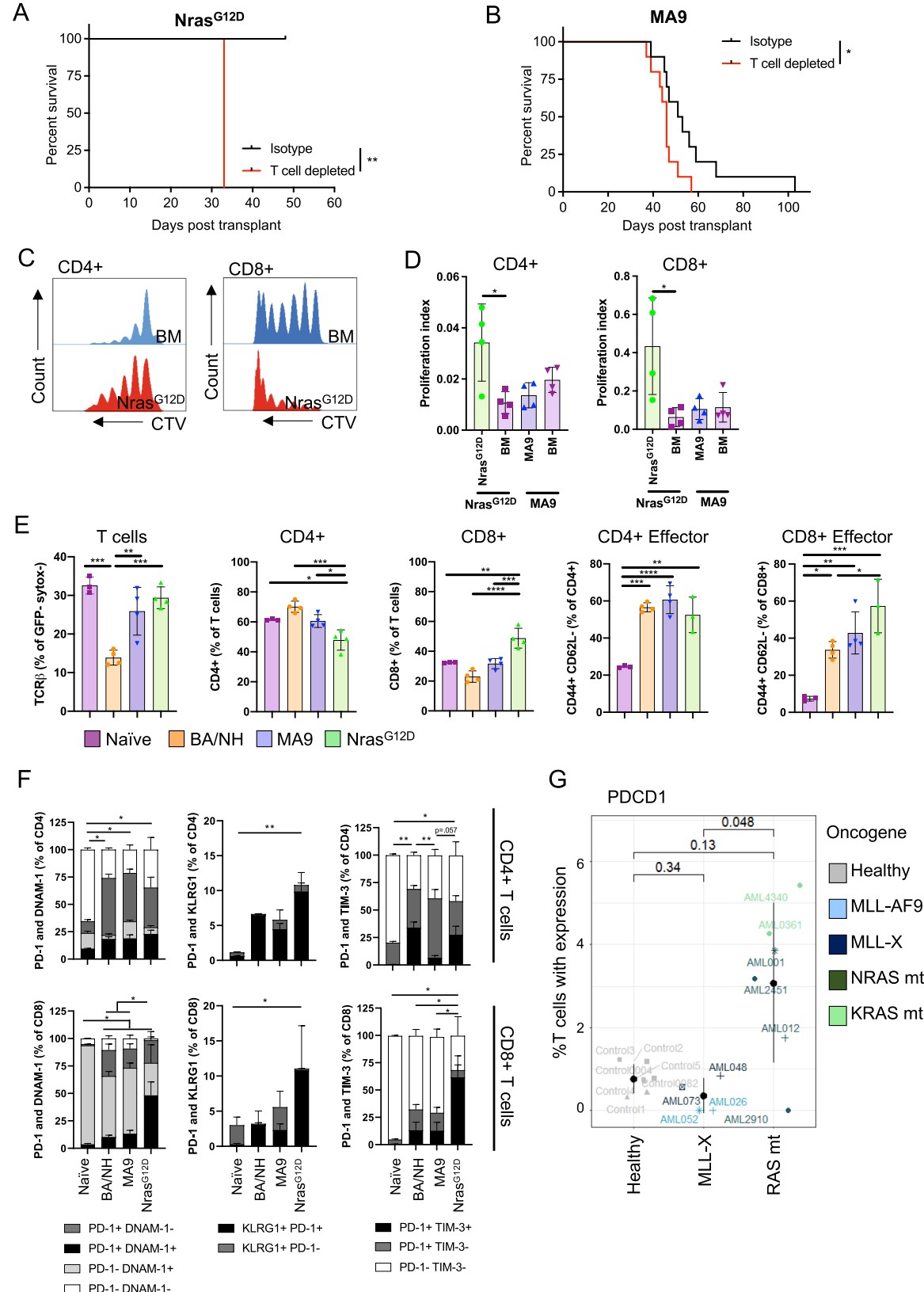

Myc expression resulted in only minor differences in MHC Class I surface expression (H2-D^b and H2-K^b), but a marked increase in PD-L1 and CD86 expression (Fig. 5E). Unexpectedly, ectopic Myc also resulted in a striking reduction in MHC Class II surface expression.

In order to functionally assess the effect of ectopic Myc expression on the anti-AML immune response in vivo, we transplanted both the Nras^G12D/Myc and Nras^G12D/EV AMLs into both immunodeficient and immunocompetent recipients (Fig. 5F). Ectopic Myc expression resulted in slightly accelerated disease progression in the Rag2^−/−γc^−/− recipients, and this difference was dramatically increased in immunocompetent recipients. Combined, this data supports the conclusion that increased

**Fig. 3 | Oncogene specificity influences the type of immune response to AML cells.** Kaplan–Meier plot comparing survival of wildtype (WT) recipients depleted of T cells and isotype control mice when transplanted with either **A** Nras[G12D] (Isotype $n = 5$, T cell depleted $n = 4$, $p = 0.0047$) or **B** MA9 AML (Isotype $n = 10$, T cell depleted $n = 10$, $p = 0.0384$) passaged through Rag2[−/−]γc[−/−] mice. **C** T cells from mice transplanted with Nras[G12D] or MA9 AML. Histograms comparing CD4[+] and CD8[+] T cell proliferation through loss of cell trace violet when cultured with or without irradiated Nras[G12D] cells isolated from Rag2[−/−]γc[−/−] recipients. **D** Proliferation index of CD4[+] ($p = 0.024$ (Nras[G12D] vs BM) or CD8[+] ($p = 0.0286$ (Nras[G12D] vs BM) T cells (as in **C**, $n = 4$ per AML genotype) after incubation with irradiated Nras[G12D], MA9 cells or untransformed BM, all isolated from Rag2[−/−]γc[−/−] recipients. Representative data from replicate experiments. Each point represents a biologically independent animal. Data are presented as mean values +/− SD. **E** Frequency of T cells (as a % of sytox- (alive) GFP- cells) ($n = 3$ (Naïve), $n = 4$ (BA/NH, MA9, Nras[G12D]), $p = 0.0045$ (BA/ NH vs MA9), $p = 0.0006$ (BA/NH vs Nras[G12D]), $p = 0.0002$ (BA/NH vs Naïve)), CD4[+] ($n = 3$ (Naïve), $n = 4$ (BA/NH, MA9, Nras[G12D]), $p = 0.0002$ (BA/NH vs Nras[G12D]), $p = 0.0118$ (MA9 vs Nras[G12D]), $p = 0.0122$ (Nras[G12D] vs Naïve)) and CD8[+] ($n = 3$ (Naïve), $n = 4$ (BA/NH, MA9, Nras[G12D]), $p = 0.0009$ (MA9 vs Nras[G12D]), $p = 0.0025$ (Nras[G12D] vs Naïve)) T cells (as a proportion of total T cells), and CD4[+] effector ($n = 3$ (Naïve, Nras[G12D]), $n = 4$ (BA/NH, MA9), $p = 0.0002$ (Naïve vs BA/NH), $p = 0.0011$ (Naïve vs Nras[G12D])) and CD8[+] effector ($n = 3$ (Naïve, Nras[G12D]), $n = 4$ (BA/NH, MA9), $p = 0.0183$ (Naïve vs BA/NH), $p = 0.0026$ (Nave vs MA9), $p = 0.0003$ (Naïve vs Nras[G12D]), $p = 0.0328$ (BA/NH vs Nras[G12D])) T cells (as a % of either CD4[+] or CD8[+] T cells) in naïve WT mice and secondary WT recipients of AML previously passaged through Rag2[−/−]γc[−/−] mice. Each point represents a biologically independent animal. Data are presented as mean values +/− SD. **F** Percentage of CD4[+] and CD8[+] T cells co-expressing PD-1, DNAM-1 (CD4[+]: $p = 0.011$ (Naïve vs BA/NH), $p = 0.0359$ (Naïve vs MA9), $p = 0.0193$ (Naïve vs Nras[G12D]); CD8[+]: $p = 0.0108$ (Naïve vs BA/NH), $p = 0.0351$ (Naïve vs MA9), $p = 0.0206$ (Naïve vs Nras[G12D]), $p = 0.0329$ (BA/NH vs Nras[G12D]), $p = 0.0438$ (MA9 vs Nras[G12D])), KLRG1 (CD4[+]: $p = 0.0076$ (Naïve vs Nras[G12D]); CD8[+]: $p = 0.0107$ (Naïve vs Nras[G12D])) and TIM-3 (CD4[+]: $p = 0.0039$ (Naïve vs BA/NH), $p = 0.0262$ (Naïve vs Nras[G12D]), $p = 0.0028$ (BA/NH vs MA9); CD8[+]: $p = 0.0317$ (Naïve vs Nras[G12D]), $p = 0.0423$ (BA/NH vs Nras[G12D]), $p = 0.0419$ (MA9 vs Nras[G12D])) from spleens of naïve WT mice and AML recipients ($n = 3–4$ per condition). Each point represents a biologically independent animal. Data are presented as mean values +/− SD. Statistics are only displayed for double positive. **G** Percentage of T cells expressing PDCD1 (PD-1) in the BM of healthy individuals ($n = 7$) or patients with MLL-translocated ($n = 4$) or mutant RAS ($n = 6$) AML. Data are presented as mean values +/− SD. Mantel-Cox test for comparison of Kaplan–Meier curves (**A**, **B**). Unpaired two-tailed $t$-test (**D**: CD4[+]), two-tailed Mann–Whitney test (**D**: CD8[+]), One-way ANOVA with Tukey's $p$-value adjustment (**E**, **G**), Welch ANOVA with Dunnett's T3 multiple comparisons test (**F**: PD-1[+]/DNAM-1[+], PD-1[+]/TIM-3[+]), Kruskal–Wallis test with Dunn's multiple comparisons test (**F**: PD-1[+]/KLRG1[+]). *$p < 0.05$, **$p < 0.01$, ***$p < 0.001$, ****$p < 0.0001$. Source data are provided as a Source Data file.

expression of Myc results in reduced immunogenicity in Nras[G12D]-driven AML.

## Discussion

The role of the host immune system in controlling certain cancers is well established, but data are lacking in the context of AML. Here, we have used genetically distinct murine models of AML to investigate oncogene-dependent immunogenicity of AML cells. These included BA/NH, a myeloid blast-crisis model with poor prognosis; MA9, aberrant activity of epigenetic modifier; and, AE/Nras[G12D] and Npm1[mutant]/Nras[G12D] AML, the latter two genotypes are associated with a favorable prognosis after treatment. We present evidence that the immunogenicity of AML cells and the quality of the immune response are directed by the specific oncogenic driver that induces the AML.

Of the oncogenic drivers tested, the strongest immune response was induced by Nras[G12D], evidenced by the greatest difference in disease latency between immunodeficient and immunocompetent mice. We found distinct expression levels of antigen presentation machinery and inhibitory and activating immune cell ligands on the different AMLs in the absence of exposure to a competent immune system. Furthermore, we found that immune cell ligand expression corresponds to the observed difference in AML immunogenicity, indicating that oncogenes influence the inherent potential of an AML cell to interact with the immune system. Specifically, BA/NH cells had low expression of MHC Class I (H2-D[b] and H2-K[b]) together with high expression of inhibitory immune checkpoint ligands PD-L1, GAL-9 and CD155 compared to MA9 and Nras[G12D]. This indicates that the non-immunogenic phenotype of BA/NH may be driven by inherent low antigen presentation and high expression of immune suppressive checkpoint molecules. In contrast, Nras[G12D] cells displayed a different cell surface phenotype with increased expression of antigen presentation machinery and immune-stimulatory ligands. Indeed, when we looked across a more extensive panel of molecular subtypes, RAS mutant human AML demonstrates high HLA Class II expression. It must be noted that MA9 AML exhibited an appreciable immunogenic phenotype despite exhibiting low levels of MHC Class I and II expression, suggesting a possible role for other immune cells such as NK cells in the control of this leukemia.

Expression of the HLA Class II presentation machinery has recently been suggested as a determinant of immune evasion in hematological malignancies[37]. It has been shown that HLA Class II expression in AML is in part dependent on the methylation status of the promoter of the transcriptional coactivator CIITA, and its expression, which is IFNγ responsive[37]. Interestingly, the expression of the co-stimulatory molecule CD80 is also IFNγ responsive[38] and mutant Ras has recently been shown to drive a cell-intrinsic interferon response via increased expression of transposable elements[39] providing a possible mechanism for the unique surface immunophenotype of the immunogenic Nras[G12D] AML. This also poses the interesting hypothesis as to whether AML immunogenicity can be enhanced by the administration of IFNγ, either alone or in combination with hypomethylating agents.

Consistent with the hypothesis that cell intrinsic differences in AML surface expression of immunomodulatory ligands dictate the potential for an anti-leukemia immune response, individual genetic aberrations also dictate the composition of the tumor microenvironment, including the degree of myeloid and lymphocyte cell infiltration and dysfunction[40]. We identified distinct oncogene-dependent immune microenvironments in AML recipient immunocompetent mice. Expansion of CD8[+] effector T cells in Nras[G12D] recipients indicates activation of the adaptive immune response that was observed to a much lesser extent in BA/NH or MA9 recipients. This was consistent with depletion experiments demonstrating a more dominant role for T cells in control of Nras[G12D] in comparison to MA9 AML. Furthermore, a higher frequency of CD8[+] T cells in the Nras[G12D] AML microenvironment of moribund mice displayed expression of immunosuppressive receptors PD-1 and TIM-3. Consistent with this, examination of T cells from AML patients at diagnosis also revealed that a higher frequency of T cells in the RAS mutant AML tumor microenvironment display gene expression of PD-1 in comparison to MLL translocated AML, suggesting that RAS mutant AML can only progress in the context of inhibitory receptor expression on T cells.

In addition to immune dysfunction, immune evasion may also be facilitated by immunoselection against the most immunogenic tumor cells, a dynamic process referred to as immunoediting[26,27] and can be mediated by T cell-dependent selection against immunogenic neo-antigens and IFNγ associated genetic instability[41–43]. Here, the more immunogenic Nras[G12D] AML showed functional immunoediting in immunocompetent mice whereas the less immunogenic MA9 AML did not, possibly reflecting the difference in the efficacy of T cells in controlling these AMLs. Immunoediting has been described in mouse sarcoma models[44], but has not been previously demonstrated in a syngeneic AML model. The different capacities for immunoediting presumably reflect the strength of the effector immune cell response and the degree of in vivo selective

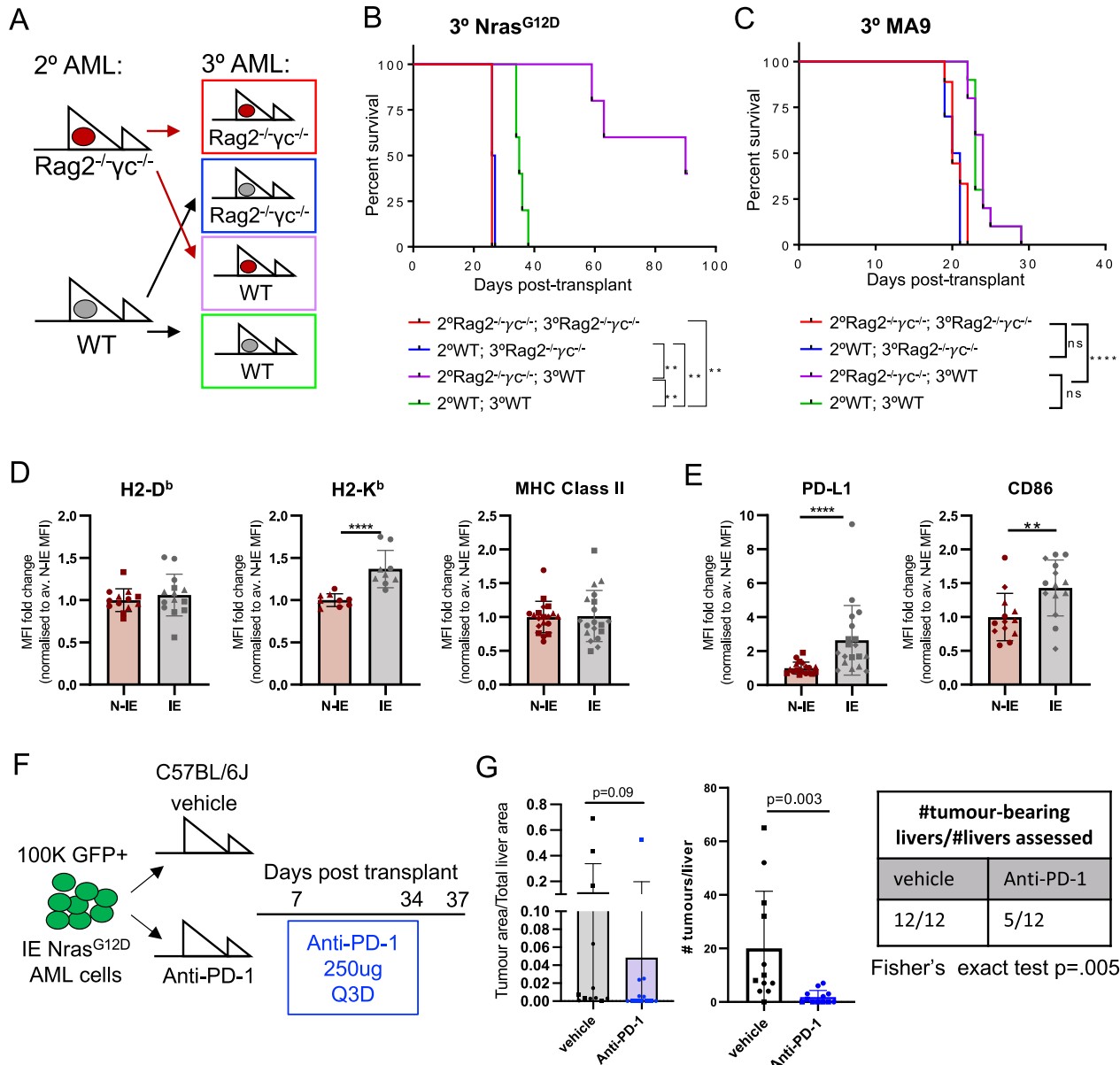

**Fig. 4 | Nras^{G12D} AML cells escape immunologic control through immunoedit-ing. A** Experimental schema for cross-over experiment to test for immunoediting. **B** Kaplan–Meier plot comparing survival of tertiary recipients transplanted with 2° Nras^{G12D} passaged through either Rag2^{−/−}γc^{−/−} or WT mice (*n* = 5 per group) (*p* = 0.0039 (2°WT;3°Rag2^{−/−}γc^{−/−} vs 2°WT;3°WT), *p* = 0.0039 (2°WT;3°Rag2^{−/−}γc^{−/−} vs 2°Rag2^{−/−}γc^{−/−};3°WT), *p* = 0.0017 (2°Rag2^{−/−}γc^{−/−};3°WT vs 2°WT;3°WT), *p* = 0.0047 (2°Rag2^{−/−}γc^{−/−};3°Rag2^{−/−}γc^{−/−} vs 2°WT;3°WT)). Representative data shown from two repeat experiments. **C** Kaplan–Meier plot comparing survival of tertiary recipients transplanted with 2° MA9 passaged through either Rag2^{−/−}γc^{−/−} or WT mice (*n* = 5 per group). Representative data shown from two repeat experiments. **D** Flow cytometry analysis of cell surface expression of immune modulatory molecules on N-IE and IE GFP⁺ Nras^{G12D} cells. MFI fold change normalized to average N-IE MFI analyzing H-2D^b (N-IE *n* = 13, IE *n* = 14), H-2K^b (N-IE *n* = 9, IE *n* = 9), MHC Class II (N-IE *n* = 19, IE *n* = 19), **E** PD-L1 (N-IE *n* = 19, IE *n* = 19) and CD86 (N-IE *n* = 13, IE *n* = 14,

*p* = 0.0090). Data are presented as mean values +/− SD, pooled data from two experiments. **F** Experimental schema for anti-PD-1 treatment of IE Nras^{G12D} recipients. Mice were transplanted with IE Nras^{G12D} on day 0. Treatment with vehicle or anti-PD-1 (250 μg per recipient) was commenced on day 7 post-transplant and continued every 3 days until day 35. **G** The proportion of tumor in a single, matched liver lobe (left), the number of tumors in a single, matched liver lobe, as quantified from H&E staining (middle; *p* = 0.003) and the number of mice with tumor in a single, matched liver lobe (table) of vehicle (*n* = 12) or anti-PD-1 (*n* = 12) treated mice pooled from two independent experiments. Data are presented as mean values +/− SD. Two-tailed Mann–Whitney test for comparison between two groups (**D**, **E**, *G*) and Mantel-Cox test for comparison of Kaplan–Meier curves, unadjusted *p*-values (**B**, **C**). \**p* < 0.05, \*\**p* < 0.01, \*\*\**p* < 0.001, \*\*\*\**p* < 0.0001. Source data are provided as a Source Data file.

pressure. Immunoediting is also seen in highly selective clinical settings, specifically after the use of CD20-directed monoclonal antibodies and CD19-directed CAR-T cell therapy for ALL where CD20 and CD19 negative relapses are observed and facilitate escape from CAR-T cell killing[45,46]. Further evidence of immunoediting is seen in AML relapse after allogeneic BM transplantation which is

frequently associated with downregulation of HLA Class II molecules[11,47] or polymorphisms in the HLA region[48].

Although immunoediting was characterized by the upregulation of the immunosuppressive ligand PD-L1, anti-PD1 therapy had limited efficacy in restoring the anti-leukemia immune response in this model, suggesting that the strategy employed in the immunoedited Nras^{G12D}

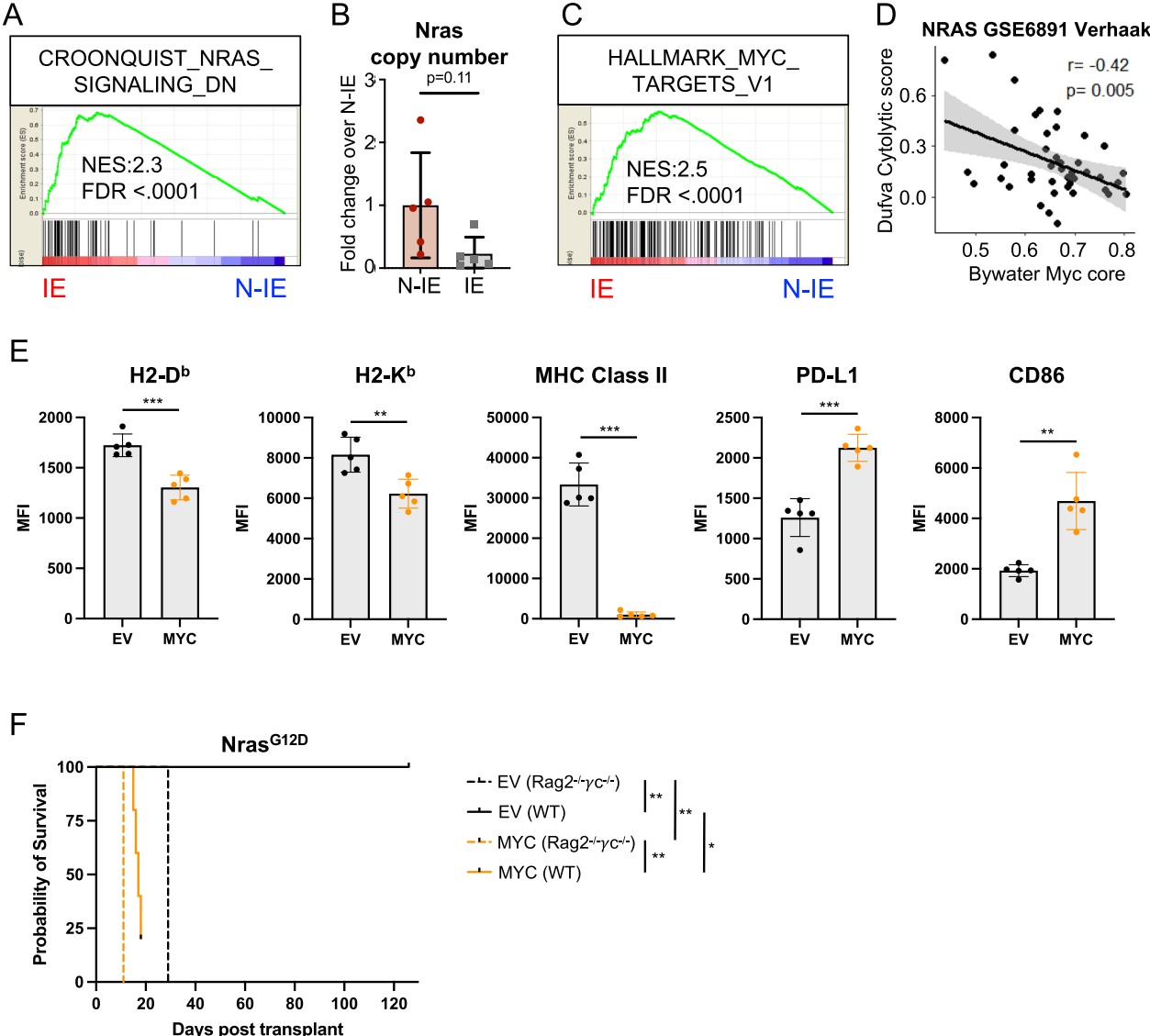

**Fig. 5 | Myc activation reduces immunogenicity in Nras^G12D-driven AML.**
**A** Enrichment of genes correlating with down-regulation of Nras signaling in immunoedited (IE) Nras^G12D AML, as determined from RNA-sequencing of GFP⁺ AML cells isolated from either immunocompetent WT (IE, *n* = 5) or immunodeficient Rag2⁻/⁻γc⁻/⁻ (N-IE, *n* = 5) recipients. **B** Relative quantification of Nras copy number in genomic DNA by qPCR in N-IE (*n* = 5) and IE (*n* = 5) Nras^G12D AML cells, expressed fold change to the N-IE mean. Each point represents a biologically independent animal, data are presented as mean values +/− SD. **C** Enrichment of genes correlating with upregulation of MYC transcriptional targets in IE Nras^G12D AML. **D** Correlation between the relative enrichment of a gene set containing core MYC transcriptional targets[36] and genes associated with cytolytic infiltrate in AML[37], using bulk expression data from NRAS mutant AML patients[29]. **E** Flow cytometry analysis of cell surface expression of immune modulatory molecules on N-IE Nras^G12D AML cells transduced with either a Myc expression construct (MYC) or empty vector (EV) and passaged in Rag2⁻/⁻γc⁻/⁻ recipients (mCherry⁺ GFP⁺ CD11b⁺ splenocytes from 5 independent Rag2⁻/⁻γc⁻/⁻ recipients) (*p* = 0.0005 (H2-D^b), *p* = 0.005 (H2-K^b), *p* = 0.0001 (MHC Class II), *p* = 0.0002 (PD-L1), *p* = 0.0079 (CD86)). Data are presented as mean values +/− SD. **F** Survival of Rag2⁻/⁻γc⁻/⁻ and WT secondary recipients transplanted with 60,000 N-IE Nras^G12D/EV or MYC AML cells. Mantel-Cox test for comparison of Kaplan−Meier curves. (*n* = 5 per group, *p* = 0.0027 (EV Rag2⁻/⁻γc⁻/⁻ vs EV WT), *p* = 0.0027 (MYC Rag2⁻/⁻γc⁻/⁻ vs MYC WT), *p* = 0.0027 (EV Rag2⁻/⁻γc⁻/⁻ vs MYC Rag2⁻/⁻γc⁻/⁻), *p* = 0.0133 (EV WT vs MYC WT)) (**F**). Unpaired two-tailed *t*-test (**E**) with Welch's correction (**B**). Two-tailed Pearson's test for association (**D**). \**p* < 0.05, \*\**p* < 0.01, \*\*\**p* < 0.001, \*\*\*\**p* < 0.0001. Source data are provided as a Source Data file.

AML to evade the immune system is likely to be multifaceted. While ICB clinical trials are ongoing in AML patients, preliminary data suggests that single agent ICB also has minimal activity in AML patients whereas ICB in combination with hypomethylating agent azacitidine has shown positive responses in a proportion of AML patients with higher pretherapy bone marrow CD3⁺ T cells and the presence of ASXL1 mutation, again suggesting that patient genetic profiles are important predeterminants of treatment efficacy[20,49].

We hypothesize that immunoediting in Nras^G12D AML is driven by the coordinate downregulation of mutant Nras expression and upregulation of Myc-driven transcription. The down regulation of mutant Nras expression is consistent with the immunogenicity of this AML being in part driven by the presentation of immunogenic neo-antigens generated from mutant Nras. Our data are supported by previous studies in mouse models of sarcoma, that identified T cell-dependent immunoselection driving immunoediting against highly antigenic neo-epitopes derived from specific genetic mutations[41,42]. KRAS mutant-specific T cells have been found in melanoma patients showing that the immune system is capable of directly targeting mutant RAS derived neo-antigens[50] and the presence of NRAS

mutations may confer better response rates to immune checkpoint therapy[51]. RAS mutant AML has a relatively favorable prognosis after treatment with chemotherapy. It could be hypothesized that this may relate to the presence of a host immune response, and that this response can be unmasked after tumor de-bulking through chemotherapy.

There is increasing evidence showing that MYC is a critical modulator of an immunosuppressive tumor microenvironment and that MYC inactivation enables recruitment of lymphocytes into tumors[32–35]. Consistent with this, we found that MYC transcriptional activity AML inversely correlates with the presence of a cytolytic immune infiltrate. We have demonstrated that the ectopic expression of Myc alone is able to reduce surface expression of both MHC Class I and II in non-immunoedited Nras$^{G12D}$ AML, in addition to driving the increased surface expression of PD-L1 and CD86. MYC activity has previously been implicated in regulating both transcription and translation of PD-L1[35,52]. However, we believe that Myc activity has not previously been linked to the expression of antigen presentation machinery. Interestingly, recent studies have described a direct role for Myc in the repression of interferon response genes in Nras$^{G12D}$ driven models of pancreatic ductal adenocarcinoma and triple negative breast cancer[53,54]. It is tempting to speculate that MYC-mediated suppression of mutant RAS driven activation of interferon signaling may be an unappreciated aspect of this cooperative oncogenic relationship that is exploited by cancers of all cell lineages.

In summary, these data point to anti-leukemic immune responses being determined by specific oncogenic profiles. Critically, in the case of Nras$^{G12D}$, extrinsic immune pressure enables outgrowth of less immunogenic AML cells. Nras$^{G12D}$ shows transcriptional plasticity under immune selective pressure, and may undergo changes in gene and surface marker expression resulting in upregulation of an immunosuppressive phenotype. As immunotherapy approaches undergo clinical testing in myeloid blood cancers, these findings highlight that personalized and targeted treatment plans may be designed according to the genetics of AML patients.

## Methods

All animal experiments were approved by the QIMR Berghofer institutional ethics committee under protocols A11605M, A1212-619M and A1212-620M.

### Murine AML models

Wild-type C57BL/6J mice were purchased from ARC Animal Resources Centre or Walter and Eliza Hall Institute for Medical Research. Rag2$^{-/-}$γc$^{-/-}$(Rag2$^{-/-}$ Il2rg$^{-/-}$) were back-crossed onto C57BL/6J. Pathogen-free mice were maintained with approval by QIMR Berghofer institutional ethics committee under protocol A11605M, A1212-619M and A1212-620M in a facility with an 8 to 8 light/dark cycle, temperature range of 19-21 degrees Celsius and 55-65% humidity. Npm1c/Nras$^{G12D}$ AML cells, generated as previously described[31], were obtained from Prof. Wallace Langdon (UWA) and Prof. George Vassiliou (Wellcome Sanger Institute).

AML from primary hematopoietic stem and progenitor cells (HSPC) were generated as previously described[55–57]. Plasmids pMSCV-MLL-AF9-IRES-GFP (pMIG-MA9), pMSCV-NUP98-HOXA9-IRES-GFP (pMIG-NH9), pMSCV-BCR-ABL-IRES-GFP (pMIG-BA), pMSCV-AML1-ETO-IRES-GFP (pMIG-AE), pMSCV-IRES-GFP and the packaging plasmid pCL-Eco were a gift from Dr. D.G. Gilliland (Boston, MA). pMSCV-GFP-IRES-Nras$^{G12D}$ (pMIG-Nras$^{G12D}$) was a gift from A/Prof Ross Dickins (Melbourne, Australia). pMSCV-Myc-IRES-mCherry was a gift from Dr Gretchen Poortinga (Melbourne, Australia). Sanger sequencing was used to determine the exact sequence over the breakpoint of each fusion protein encoded in the listed constructs (Supplementary Table 1). Plasmid DNA was isolated from transformed E.coli using PureLink™ HiPure Plasmid Filter Maxiprep Kit (ThermoFisher) according to the manufacturer's instructions, with specific oncogenic

sequences confirmed using Sanger sequencing. Retroviruses were packaged by co-transfection of plasmids using FuGENE (Roche) into HEK293T cells (ATCC, CRL-11268).

Bone marrow (BM) from Rag2$^{-/-}$γc$^{-/-}$ mice (8-12 weeks of age) was depleted using Ter119, Gr1, and CD11b biotin-conjugated antibodies and Dynabeads (Invitrogen) according to the manufacturer's instructions. Lineage-depleted BM cells at $2 \times 10^6$ cells/mL were cultured overnight (37 °C, 5% CO$_2$) in RPMI supplemented with 10% FCS, 100 IU/mL Pencillin/Streptomycin, 10 ng/mL murine recombinant IL-3 (Peprotech), murine recombinant IL-6 (Peprotech) and 50 ng/mL murine recombinant stem cell factor (SCF) (Peprotech). Cells were re-suspended with 1 mL of unconcentrated retrovirus per well with polybrene (8 µg/mL) and HEPES (30 µL/mL) and mIL-3 10 ng/ml, mIL-6 10 ng/ml and mSCF 50 ng/ml (Peprotech). Transduction combinations included: MLL-AF9 alone; BCR-ABL and NUP98-HOXA9; or AML1-ETO and Nras$^{G12D}$ expressed using the retroviral backbone MSCV-IRES-GFP.

BM cells were transduced by two room temperature spin infections for 90 mins at 3000 rpm separated by three-hour incubation at 37°C, 5% CO$_2$. For primary (1°) transplants, BM cells were assessed for viability and GFP expression by flow cytometry prior to transplantation by lateral tail vein injection into primary Rag2$^{-/-}$γc$^{-/-}$ recipients. For secondary (2°) and tertiary (3°) transplants, equal numbers of GFP$^+$ AML cells were transplanted into non-irradiated wild-type C57BL/6J or Rag2$^{-/-}$γc$^{-/-}$ recipients (150 K per recipient, groups matched for gender). Mice recruited to survival studies were euthanised when ethically required by a cumulative clinical score based on weight loss, posture, activity and white cell count.

Cryopreserved 1° NrasG12D AML-bearing splenocytes were recovered and immediately transduced with either pMSCV-Myc-IRES-mCherry or the empty vector backbone retrovirus in short-term culture, as detailed above. Double positive GFP$^+$, mCherry$^+$ cells were sorted (BD FACSAria™) and expanded via transplantation in Rag2$^{-/-}$γc$^{-/-}$ recipients.

### Genotyping of murine AML models

Genomic DNA was isolated from sorted GFP$^+$ cells using Quick-Extract™ DNA Extraction Solution (Illumina) according to the manufacturer's instruction. PCR was performed using the following primers: NUP98-HOXA9 (Fwd-5′gcacaaataccagtgggaata 3′, Rev-5′gggcaccgc tttttccgagtg 3′, 373 bp product), BCR-ABL (Fwd-5′cagatgctgaccaa ctcgtgt 3′, Rev-5′gtttgggcttcacaccattcc 3′, 377 bp product), AML1-ETO (Fwd-5′gagggaaaagcttcactctg 3′, Rev-5′gaaggcccattgctgaagc 3′, 325 bp product), NRas$^{G12D}$ ORF (Fwd-5′ccagtacatgaggacaggcg 3′, Rev-5′ acttgttgcctaccagcacc 3′, 145 bp product)(Fwd-5′ggacacagctggacaa gagg 3′, Rev-5′cacacttgttgcctaccagc 3′, 190 bp product).

### Histology

Histology samples were processed by the QIMR Berghofer Histology Facility. Briefly tissues were fixed in 10% neutral buffered formalin, embedded in paraffin prior to staining with haematoxylin Aarmstadt Hx Crystals (Merck) and Eosin Y (H&E). In addition, tissues were stained for peroxidase labeled Ki67 (SP6, ab16667, Abcam) and phosphorylated histone H3 (p-H3) (polyclonal, 06-570, Merck Millipore) with images captured on a Nikon Eclipse Ci, DS-Fi2 microscope or the Aperio Scanscope XT using Scanscope (version 102.0.7.5).

### In vivo antibody experiments

In vivo antibody depletion in wild-type mice was performed using the following antibodies: anti-CD4 (100 µg, GK1.5; Bio-X-Cell BE0003-1), anti-CD8β (100 µg, 53.5.8; Bio-X-Cell BE0223) and control IgG (100 µg, HRPN; Bio-X-Cell BE0088). Antibodies were injected into the intra-peritoneal cavity on days −1 and 0 and then weekly for the duration of the experiment. Anti-PD-1 (250 µg, RMP1-14, Bio-X-Cell BE0146) immune checkpoint inhibitor experiments commenced 7 days after

transplant, with treatment every 3–4 days, with a total of 9 doses administered.

## T cell proliferation assay

Non-irradiated wild-type C57BL/6J mice were transplanted with Nras$^{G12D}$ or MA9 AML previously expanded in Rag2$^{-/-}$γc$^{-/-}$ mice. Whole splenocytes were harvested, labeled with Cell Trace Violet (CTV) and incubated for 72 h at a 5:1 ratio with and without irradiated (40 Gy) Nras$^{G12D}$, MA9 (passaged through Rag2$^{-/-}$γc$^{-/-}$ mice) or non-transformed Rag2$^{-/-}$γc$^{-/-}$ bone marrow (BM) cells and 0.01µg/mL soluble CD3 (2C11, Biolegend). Dilution of CTV on CD4$^+$ and CD8$^+$ T cell populations was evaluated by flow cytometry. The proliferation indexes of CD4$^+$ and CD8$^+$ T cells were calculated by dividing the total number of divisions by the number of cells that underwent division.

## Blood analysis

Blood collected into EDTA-coated tubes was analyzed on a Hemavet 950 analyser (Drew Scientific) using Hemavet DMS Capture (version 1.0.0) software.

## Fluorescence-activated cell sorting and analysis (FACS)

Spleens and livers were harvested into ice cold FACS buffer (PBS supplemented with 2% FCS v/v) and then emulsified through a 70 µM filter (BD Biosciences), centrifuged and washed in FACS buffer. Liver mononuclear cells were isolated using isotonic percoll gradient. Tissue samples were treated with Red Blood Cell Lysis Buffer (BD Pharmlyse, BD Biosciences). Samples were incubated in CD16/CD32 blocking antibody (clone 93; Biolegend) before staining with appropriate fluorochrome-conjugated antibodies at the listed dilution (Supplementary Table 2). Post-acquisition analyses were performed using FlowJo software V10.0 (Treestar, CA). Cell analysis and sorting were performed using the BD FACS LSR Fortessa™ or BD FACSAria™ using FACS Diva Software (version 8.0.1).

## RNA-sequencing and bioinformatics analysis

GFP positive AML cells were isolated on BD FACSAria™, washed in ice cold PBS including 1:500 protease inhibitor cocktail (Sigma Aldrich) prior to snap freezing on dry ice. Total RNA was isolated from frozen cell pellets using the Arcturus PicoPure RNA Isolation Kit (Applied Biosystems). Samples were quantitated using a Qubit RNA HS Assay Kit (Molecular Probes), with integrity confirmed using the RNA 6000 PICO Kit (Agilent Technologies) and Agilent 2100 Bioanalyser (Agilent Technologies).

Oligo d(T) captured mRNA (100 µg) was processed for Next Generation Sequencing (NGS) using the NEB Next Ultra II RNA Library Prep Kit for Illumina (New England Biolabs). Quality was assessed using the High Sensitivity DNA Kit (Agilent) on the Agilent 2100 Bioanalyser with quantification using the Qubit DNA HS Assay Kit (Molecular Probes). Final libraries were sequenced using a high output single -end 75 bp flow cell (version 2) on the Illumina Nextseq 550 platform using NextSeq System Suite (version 2.1.2). Reads were trimmed for adapter sequences using Cutadapt (version 1.11) and aligned using STAR[58] (version 2.5.2a) to the GRCm38, with assembly using the gene, transcript, and exon features of Ensembl (release 67). Expression was estimated using RSEM (version 1.2.30), with transcripts with zero read counts across all samples removed prior to analysis. Normalisation of read counts was performed by dividing by million reads mapped to generate counts per million (CPM), followed by the trimmed mean of M-values (TMM) method from the edgeR (version 3.28.1) package[59]. For the differential expression analysis, reads were filtered but not normalized, since edgeR performs normalisation (library size and RNA composition) internally. The glmFit function was used to fit a negative binomial generalised log-linear model to the read counts for each transcript. Transcript wise likelihood ratio tests were conducted for

each comparison. Log2 transformed, normalized read counts were used for heatmaps and principle component analysis (PCA).

Gene set enrichment analysis (GSEA) was performed using GSEA (version 4.1.0) from Broad Institute[60,61]. *P*-values were generated form 1000 gene set permutations, excluding gene sets with more than 3000 genes or <5 genes against custom made gene sets and Broads Hallmark and C2 database.

## Quantitative polymerase chain reaction (qPCR)

DNase free RNA was extracted from sorted GFP$^+$ Nras$^{G12D}$ cells using Arcturus PicoPure RNA Isolation Kit (Applied Biosystems) according to the manufacturer's instructions. Reverse transcription was performed using Maxima H minus first strand cDNA synthesis kit with dsDNase according to the manufacturer (Thermofisher Scientific). Genomic DNA was extracted using QIAGEN DNeasy Blood and Tissue kit according to the manufacturer's instructions. RNA and DNA quantification concentration and purity were determined by Nanodrop spectrophometer (ThermoFisher Scientific). Q-PCR primers were designed using Primer3web (http://bioinfo.ut.ee/primer3/).

Primers used: Nras: Forward 5′ TGTTGGGAAAAGCGCCTTGA 3′ and reverse 5′ CCTGTCCTCATGTACTGGTCT 3′. Beta-actin (Actβ): Forward 5′ GACGATATCGCTGCGCTGGT 3′ and reverse 5′ CCAC GATGGAGGGGAATA 3′. Primer specificity was determined by melt-curve analysis. Gene expression and copy number were quantified by Sybr green reaction (Life technologies) using ABI Viia7 qPCR machine with the QuantStudio Real-Time PCR System (version 1.3) according to standard protocols. Gene expression was determined by standard curve analysis and normalized to Actβ with relative gene expression and copy number expressed as a fold change.

## Publicly available microarray and single cell RNA-sequencing data analysis

Microarray array data with accession GSE6981 were downloaded, pre-processed and normalized as previously described[29]. Samples from AML patients with NRAS mutation (*n* = 42) and MLL-translocation (*n* = 10) were extracted for analysis. Using the GSVA (version 1.34.0) R package function ssgsea the relative enrichment of gene sets across was assessed for all genetic aberrations with more than 3 patients for genes encoding MHC Class II and for NRAS mutant samples was assessed for gene sets associated with cytolytic immune infiltration in AML[37] or Myc gene target activation[36].

Pre-processed and annotated single cell RNA-sequencing data of 5 AML patients with MLL-translocation, 6 patient with RAS mutations (2 KRAS,4 NRAS) and 7 healthy controls were obtained from GEO with accession GSE185381[30]. For MHC Class II gene expression comparison across oncogenes, CD33 expressing malignant annotated cells were extracted and B, T and NK cells excluded. For each sample MHC Class II gene expression was averaged. For comparison of the percentage of PDCD1 expressing cells oncogenes annotated T cells were extracted.

## Statistical and bioinformatics analysis

Statistical analysis was performed using Prism (v7.02) as follows: Log-rank (Mantel-Cox) test for p values for all Kaplan−Meier survival analyses. To compare two groups unpaired Student's *t*-test when normality and equal variance assumptions are met, Mann−Whitney test otherwise; unless otherwise described for more than two groups when normal distribution and equal variance assumption are met ordinary one-way ANOVA with post-Tukey multiple comparison test was performed, in case of violation of equal variance we performed Welch ANOVA or Kruskal−Wallis test with Dunn's multiple comparisons test when normality is violated. Test for association between paired samples, using Pearson's product moment correlation coefficient.

**Reporting summary**

Further information on research design is available in the Nature Portfolio Reporting Summary linked to this article.

## Data availability

The RNA sequencing datasets generated in this study have been deposited in the NCBI Gene Expression Omnibus under accession numbers GSE164951 and GSE207316. Publicly availably AML microarray data used in this study have the accession number GSE6891. Publicly available AML and healthy single cell CITE and RNAseq data have the accession number GSE185381. RNAseq reads from murine experiments were mapped to mouse genome build GRCm38 with ensembl v70 gene model downloaded from ensembl on 11 June 2015. Source data are provided with this paper. The remaining data are available within the Article, Supplementary Information or Source Data file. Source data are provided with this paper.

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

## Acknowledgements

We are grateful for the assistance of the QIMR Berghofer animal house, flow cytometry, microscopy and histology facilities and helpful comments from members of the Lane Lab and Immunology Department at QIMR Berghofer Medical Research Institute. Specifically, Mark Smyth, Rachel Kuns, Madeleine Headlam and Keyur Dave. We thank Thomas Mercher for helpful comments and review of the manuscript. We gratefully acknowledge the support of Neil Herron and the Herron Family Trust as supporters of leukaemia research in Queensland, CSL Centenary Fellowship (S.W.L.), NHMRC (Investigator Grant (S.W.L.) 1195987, Ideas Grant (M.J.B.) 2003575, Early Career Fellowship (M.J.B.) 1072477), Gordon and Jessie Gilmour Fellowship and Leukaemia Foundation of Australia (R.A.). C.B. and human AML work were supported by NHMRC Project Grant (1157263). M.B. was supported by a fellowship of the Mildred-Scheel Stiftung fur Krebsforschung (Germany). F.S.F.G. was supported by a NHMRC Early Career Fellowship (1088703), a National Breast Cancer Foundation (NBCF) Fellowship (PF-15-008).

## Author contributions

R.A., J.S., S.W.L., and M.J.B. conceptualized and designed experiments, analyzed and interpreted data. R.A. and M.J.B. wrote the manuscript. R.A., J.S., R.H., Y.J., C.B., M.W., L.C., A.P., E.C., and A.S. performed experiments and data collection. J.S. performed bioinformatical analyses. C.B., M.W., M.B., F.S.F.G., S.A.M., S.J., T.B., K.N., G.R.H., and I.A. provided critical reagents and data interpretation, and/or intellectual input and supervision of the study. All authors contributed to and edited the manuscript.

## Competing interests

S.W.L. has participated in advisory boards for Celgene, Novartis and Janssen, and has received research funding from Janssen and Celgene/Bristol Myers Squibb for unrelated projects. G.R.H. has consulted for Generon Corporation, NapaJen Pharma, iTeos Therapeutics, Neoleukin Therapeutics and has received research funding from Compass Therapeutics, Syndax Pharmaceuticals, Applied Molecular Transport, Serplus Technology, Heat Biologics, Laevoroc Oncology, Genentech and iTeos Therapeutics. F.S.F.G. is a consultant and has a funded research agreement with Biotheus Inc. The remaining authors declare no other competing interests.
