## [Peer Review File · Nature Communications]

Oncogenic drivers dictate immune control of acute myeloid leukemiaREVIEWER COMMENTS

Reviewer #1 (Remarks to the Author):

Manuscript by Austin et al entitled Oncogenic drivers dictate immune control of acute myeloid leukemia. Using distinct oncogenic models, the authors describe that specific AML oncogenes can dictate immunogenicity and immune response, which underlie immune evasion. The authors make use of retroviral murine overexpression models for NUP98-HOXA9+BCR-ABL, MLL-AF9, and AML1-ETO+NRasG12D. Transduced murine lin⁻ BM cells are propagated in immunodeficient or immunocompetent mice and transformation trajectories are studied. The authors also make use of also of a transgenic NPMc/NRas12D model. In particular AML1-ETO+NRasG12D murine leukemic cells, as well as patient cells, express relatively high levels of MHC class II molecules potentially underlying the greater impact of the immune system on the delay of leukemia onset, and AML1-ETO+NRasG12D MHCII^{-/-} cells were more efficient in inducing leukemia upon transplantation in immunocompetent mice. Leukemia cell intrinsic mechanisms also play a role since propagation of either AML1-ETO+NRasG12D or MLL-AF9 murine leukemic cells in immunocompetent mice resulted in the generation of more aggressive leukemia-inducing cells as compared to cells that were propagated in immunodeficient mice. Immunoedited cells were enriched for MYC signatures, which did not necessarily further drive intracellular proliferation programs but were rather considered to provide immune suppressive signals to the tumor microenvironment. Furthermore, clonal selection might also provide a role since immunoedited AML1-ETO+NRasG12D cells expressed reduced RAS transcriptional activity, coinciding with an upregulation of MHCII molecules. Ultimately, an increase in antigen presentation in AE/NRAS cells would result in T cell exhaustion. Overall, this is certainly interesting work describing the impact of specific oncogenes on immunogenicity and immune response making elegant use of immunodeficient and immunocompetent models. I do have a number of considerations, remarks and suggestions.

How do data described here in murine retroviral overexpression models relate to the situation in human patients? Oncogenes are overexpressed at relatively high levels, onset of leukemia is extremely fast, at least in some of the models (from 15-30 days to about 100 days in the AE/NRAS model), and the effects of the immune system on the transformation process were evaluated in 2nd transplant models. It is suggested that "...a graded immune response to AML subtypes is specified by individual oncogenes", but it also appears as if timing plays an important role, whereby the faster the onset of disease (in immunodeficient models, so independent of immune evasion and most likely related to cell intrinsic mechanisms that drive aggressive proliferation, anti-apoptosis, self-renewal and a differentiation block) the weaker the impact of the immune system. If eg NUP98-HOXA9/BCR-ABL cells were transplanted with much less cell numbers than 150k, which would result in slower onset of disease, would it not be conceivable that then also differences in disease trajectories would be observed after transplantation in Rag2^{-/-}γc^{-/-} versus C57BL/6J mice? To rule out timing effects, and to firmly establish that there are oncogene-specific mechanisms that control immunogenicity or immune evasion one would like to obtain somewhat more direct evidence showing eg how AML1-ETO, RAS, NH etc would differentially impact on eg MHCII expression.

Data in Fig 2B-C aims to put the association between oncogene expression and cell intrinsic immunogenicity in a human perspective, which is a nice addition. I do however wonder how easily these data can be generalized. For as far as I can see in various independent datasets the expression levels of HLA-DQA1 in human leukemias and normal stem/progenitor cells is quite diverse, with on average the highest expression in inv16 patients, but in fact in all genetic subgroups there appear to be patients that have either high or low expression of HLA-DQA1. And if anything, in most cases in most datasets there appear to be lower levels of HLA-DQA1 in AML compared to normal HSCs/MPPs. Regarding the single cell data taken from the van Galen paper: more details could have been provided. Which cells are actually analyzed and plotted in Fig.2s and Suppl. Fig.2C? All cells? Only the AML1-ETO translocated cells? One wonders why MHC II molecules would be upregulated in leukemic cells? This clearly would not provide any benefit. It has previously also been suggested that cell surface expression of HLA-DR,-DQ and -DP is in fact often lost on leukemia cells, due to downregulation of the HLA class II regulator CIITA. Earlier work also suggested that MHCII is epigenetically repressed in AML cells and that decitabine treatment would impair leukemogenesis by inducing more MHCII. While it is indeed true that in

that Dufva et al paper (PMID: 32649887) it was suggested that, while in the majority of AML subgroups are HLAII low, AML harboring CBFB-MYH11 or RUNX1-RUNX1T1 translocations were characterized by higher HLA II expression and CIITA hypomethylation compared to other subtypes, it would be good to explore this in somewhat more detail. It is clear that differences exist between AML subtypes with regard to the level of downregulation of the antigen-presentation response, but whether this is truly increased in AML1-ETO cells compared to normal HSPCs and what the mechanistic link between AML1-ETO, HLAII and possibly CIITA is remains unclear. Line 241: "These data reveal discrete effects of oncogenic drivers on immune regulatory molecule expression in AML cells, supporting a model whereby AE/NrasG12D AML has greater potential to interact with the immune system." Can we truly interpret this as a downstream consequence of oncogenic drivers? I think the authors could attempt to more thoroughly investigate the MHC class II landscape across multiple genetically distinct subgroups of AML patients and link this mechanistically to AML1-ETO.

Furthermore, in the last part of the paper the authors go in great detail to study the T cell landscape in human AML1-ETO patients, but whether the described early T cell exhaustion is the consequence of enhanced antigen presentation is not immediately clear.

In Figure 2F it is shown that BA/NH had the highest expression of the immunosuppressive ligands PD-L1 and GAL-9 compared to MA9 and AE/NrasG12D, but I guess the authors then conclude that these factors are not dominant since no differences were seen in the transformation process of BA/NH transplanted cells in immunodeficient versus immunocompetent models?

Also in the immunoediting experiments described in Fig.3, it is concluded that AML1-ETO+NRasG12D can undergo immunoediting when propagated in immunocompetent mice, but MLL-AF9 cells not. Again, the kinetics of leukemic onset in the MLL-AF9 model are significantly faster, and I wonder whether differences would exist when fewer MLL-AF9 cells were transplanted. Moreover, the transcriptional changes that were seen in immunoedited AML1-ETO+NRasG12D cells, were these signatures not seen in MLL-AF9 cells that were propagated in immunocompetent mice versus immunodeficient mice?

Besides retroviral overexpression models the authors also make use of a knockin NPMc/NRasG12D model, and this model obviously does not suffer from potential non-physiological high level of oncogene expression. Intriguingly, in this model also no upregulation of H-2Db and MHC Class II molecules was observed, but there an upregulation of ligands with potential immunosuppressive function, PD-L1 and CD86, was observed. What would explain the difference between this knockin model with the AML1-ETO+NRasG12D retroviral model? Does RAS signal differently in the context of AML1-ETO vs NPMc? Does the expression level of RAS matter? It would be of interest to determine whether MHCII expression levels go up upon retroviral transduction with AE, NRASG12D or both, before transplantation in mice. Line 295: "... we observed an inverse relationship between the variant allele frequency of mutant NRAS and the expression of multiple HLA (MHCII) genes, including HLA-DQA1, in NRAS-mutated human AML [41](Fig. 3H, Supplementary Fig. S3F)". Since MHCII did not go down in IE AE/NRASG12D cells one wonders about the exact molecular pathways downstream of RAS that would drive immunogenicity, or whether more indirect mechanisms would be involved.

line 341 " This indicates that NK cells contribute to immune control of MA9 and AE/NrasG12D AML, whereas T cells play a more important role in immune control of AE/NrasG12D AML." Does the data truly support this notion? Not clear from fig 4A-B, depletion of T cells still enhances tumorigenicity in the MA9 model.

Fig 4E-G: How should the reduction in T cells in the BA/NH model be interpreted since there was no difference in disease onset between immunodeficient versus immunocompetent mice?

Reviewer #2 (Remarks to the Author):

Austin et al provide a comprehensive and thorough description of how specific AML oncogenes either active or suppress the immune response. They demonstrate that specific AML oncogenes

control immunogenicity and immune escape through immunoediting. They then show that immunoediting in AML is mediated through transcriptional plasticity, modulation of immunosuppressive molecules and clonal selection. Overall the claims of the paper are supported by convincing data in human and mouse model systems. Several concepts require additional clarification and further support.

- The majority of mouse AML data is generated by the retroviral transduction/transplantation approach, which can result in preferential viral incorporation into distinct progenitor cell states. To what degree are the differential responses to immune escapes mediated by the cell of origin rather than the oncogenic driver?
- The authors suggest that the distinct responses to immune escape in the AML models are in part due to activated MYC. Can suppression of MYC rescue some of the observed phenotypes?
- One of the most interesting observations is that increased MYC signaling and down-regulation of NRAS may contribute to immune escape. The authors should test this concept by suppressing MYC and reactivating the RAS-RAF-MEK pathway in an attempt to restore immune suppression.
- BA/NH express low levels of MHC class I genes along with high expression of immune check point ligands. What is the suspected mechanism for these specific gene changes?
- Figure 1(E-G) – Is the disease manifestation in Rag2^{-/-} and C57BL6 the same or different for each of the oncogenic drivers? In other words, is the type of AML similar in the two models?
- Is there an explanation for why MA9 AML is delayed in C57BL6 (relative to Rag2^{-/-}; Figure 1F) but not for BA/NH AML (Figure 1E) even though both AMLs express similar levels of MHC genes (Figure 2)?
- The anti-PD1 data (Figure 3M) is modest and highly variable.

Reviewer #3 (Remarks to the Author):

Austin et al provide a comprehensive and high-quality manuscript. To my knowledge*, the data are technically sound, appropriately analysed and interpreted. Appropriate controls have been used. Their conclusions are sound and claims not overstated. Contextualization in introduction and discussion is excellent and accessible to a broad audience. The manuscript is very clearly written with enough detail to understand the experiments in nearly all places. They provide novel data on the relationship between oncogenic driver and immune control of AML and detailed studies to understand the mechanisms of immunogenicity and immune escape. As the authors point out, immune-directed therapies potentially do have a role in this disease, if used in combination and in the right circumstances. Therefore, this paper contains information important to advancing personalized therapy for AML

*I do not have expertise in performing mouse experiments, so cannot comment on methodology of model generation/transfer and/or strain selection.

Key results are that immunocompetent mice survived longer in MA9 but especially in AE/NRas disease. This is accompanied by MHC class II expression higher on AE/NRas than MA9 in models, AML patients at bulk tumour and single cell level. Furthermore, AE/NRas AML generated on an MHC II background had shorter survival than when generated on a WT background. They authors found accelerated disease in immunocompetent mice with AE/NRas having previously passed through an immunocompetent (vs an immunodeficient host), accompanied by increased MYC expression, reduced NRas expression and reduced mutant NRas copy number. They established that NK/T depletion accelerated disease progression in AE/NRas and that AE/NRas in co-culture stimulated T cell proliferation. Differences in tumour environment T cells were not particularly marked but increase in CD8 Effector proportion in AE/Ras was noteworthy. Alterations in immunosuppressive molecule expression was found in AE/NRas CD8 T cells. scRNAseq data on immune-enriched AML1-ETO BM added detail and supported these observations.

Specific points

1. Please make it a little clearer at the start of the results which mutations represent which risk category. Is there any reason these models of AML were chosen besides fitting into favourable, intermediate and adverse prognostic subclasses? Specifically, NUP98-HOXA9 is a rare translocation and I was curious why this was selected.

2. Why do you think it is that primary MA9 and AE/NRas mice survive 40-80 days post-transplant (Fig S1B) but secondary mice also on a Rag2^{-/-}γc^{-/-} background survive 20 days (Fig 1F)?

3. Sup Figure 2. In understand from Fig S1 that 11b expression should capture most of the myeloblasts. However the % of 11b⁺ cells in BA/NH in Fig S2A seems very low. To understand this it would be helpful to clarify in the legend what tissue is being sampled and at what timepoint following transfer? Are you sure the CD11b gate is comprehensively sampling BA/NH blasts? Sup Figure 2D How many times performed?

4. I cannot locate the figure panel providing evidence that there is no intrinsic difference in proliferative capacity associated with IE. We are referred to Fig 3B but that is a survival curve.

5. The authors use a creative way to look at correlation between MYC overexpression and immune environment, but it requires more explanation and the correlation is not wholly convincing. It is clear where the score comes from (ref 39). Where is the AML1-ETO data from? What is it- BM?. How is Myc measured here?

6. A visual representation of the overall findings would be a worthwhile addition

RESPONSE TO REVIEWERS' COMMENTS

Reviewer #1 (Remarks to the Author):

Manuscript by Austin et al entitled Oncogenic drivers dictate immune control of acute myeloid leukemia. Using distinct oncogenic models, the authors describe that specific AML oncogenes can dictate immunogenicity and immune response, which underlie immune evasion. The authors make use of retroviral murine overexpression models for NUP98-HOXA9+BCR-ABL, MLL-AF9, and AML1-ETO+NRasG12D. Transduced murine lin- BM cells are propagated in immunodeficient or immunocompetent mice and transformation trajectories are studied. The authors also make use of also of a transgenic NPMc/NRas12D model. In particular AML1-ETO+NRasG12D murine leukemic cells, as well as patient cells, express relatively high levels of MHC class II molecules potentially underlying the greater impact of the immune system on the delay of leukemia onset, and AML1-ETO+NRasG12D MHCII^{-/-} cells were more efficient in inducing leukemia upon transplantation in immunocompetent mice. Leukemia cell intrinsic mechanisms also play a role since propagation of either AML1-ETO+NRasG12D or MLL-AF9 murine leukemic cells in immunocompetent mice resulted in the generation of more aggressive leukemia-inducing cells as compared to cells that were propagated in immunodeficient mice. Immunoedited cells were enriched for MYC signatures, which did not necessarily further drive intracellular proliferation programs but were rather considered to provide immune suppressive signals to the tumor microenvironment. Furthermore, clonal selection might also provide a role since immunoedited AML1-ETO+NRasG12D cells expressed reduced RAS transcriptional activity, coinciding with an upregulation of MHCII molecules. Ultimately, an increase in antigen presentation in AE/NRAS cells would result in T cell exhaustion. Overall, this is certainly interesting work describing the impact of specific oncogenes on immunogenicity and immune response making elegant use of immunodeficient and immunocompetent models. I do have a number of considerations, remarks and suggestions.

Q1. How do data described here in murine retroviral overexpression models relate to the situation in human patients? Oncogenes are overexpressed at relatively high levels, onset of leukemia is extremely fast, at least in some of the models (from 15-30 days to about 100 days in the AE/NRAS model), and the effects of the immune system on the transformation process were evaluated in 2nd transplant models. It is suggested that "...a graded immune response to AML subtypes is specified by individual oncogenes", but it also appears as if timing plays an important role, whereby the faster the onset of disease (in immunodeficient models, so independent of immune evasion and most likely related to cell intrinsic mechanisms that drive aggressive proliferation, anti-apoptosis, self-renewal and a differentiation block) the weaker the impact of the immune system. If eg NUP98-HOXA9/BCR-ABL cells were transplanted with much less cell numbers than 150k, which would result in slower onset of disease, would it not be conceivable that then also differences in disease trajectories would be observed after transplantation in Rag2^{-/-}γc^{-/-} versus C57BL/6J mice? To rule out timing effects, and to firmly establish that there are oncogene-specific mechanisms that control immunogenicity or immune evasion one would like to obtain somewhat more direct evidence showing eg how AML1-ETO, RAS, NH etc would differentially impact on eg MHCII expression.

Thanks for this interesting point regarding how cell intrinsic differences in disease kinetics may affect the ability to determine a difference in the strength of the anti-AML immune response across models. Although cell intrinsic disease kinetics could influence the magnitude of the delay in disease latency observed between Rag2^{-/-}γc^{-/-} and wild type recipients, the existence of any significant difference in disease latency between immunodeficient and immunocompetent recipients still allows the conclusion that the presence of a competent immune system alters disease progression. The question then becomes whether the magnitude of the difference can be quantitatively compared across models. We believe this is still a valid comparison with models that have a similar disease latency in Rag2^{-/-}γc^{-/-} recipients, specifically the MA9 and AE/Nras^{G12D} models, with a 17-20day and 21-23days latency respectively. To address the Reviewer's suggestion that performing serial dilution transplantation in the BA/NH model would allow us to further confirm that disease kinetics does not influence immunogenicity in this disease

genotype, we have now performed limiting dilution transplantation of BA/NH leukemias in Rag2^{-/-}γc^{-/-} recipients to achieve a similar latency to that observed in the AE/Nras^{G12D} model. Informed by this study we used a limiting cell dose (1k cells) to examine the impact of a competent immune on disease latency via comparative transplant into both Rag2^{-/-}γc^{-/-} and wild type recipients. We have confirmed that transplantation of BA/NH AML into immunocompetent recipients failed to increase disease latency, with a significant decrease in latency observed, possibly due to preferential engraftment in wild type spleen (Rag2^{-/-}γc^{-/-} mice have atrophic spleens). This data, provided below for easy reference, has now been included in the revised manuscript (Supplementary Fig. 1H)

We agree with the reviewer that it would be informative to obtain evidence as to whether the oncogenes investigated can drive changes in immunogenic ligand expression directly. In order to address this we performed flow cytometry for immunogenic ligand expression on primary HSPCs 72hrs post retroviral transduction. Short-term expression of Nras^{G12D} was able to significantly upregulate MHC Class II, CD86 and PD-L1 expression, indicating a direct role of Nras^{G12D} in driving the high levels of expression of these ligands observed in the transformed Nras^{G12D}-driven disease. In support of this finding, mutant Ras has been shown to driven interferon signalling via the de-repression of transposable elements, a pathway that is known to drive increased expression of MHC Class II and CD80 (1). In contrast, short-term expression of neither BCR-ABL nor NUP98-HOXA9 alone was able to drive the high levels of CD80, GAL-9 or CD155 observed in the transformed BA/NH AML. This data has now been included in the revised manuscript (Supplementary Fig. 2D and 3B). Thanks again for this excellent suggestion.

Q2. Data in Fig 2B-C aims to put the association between oncogene expression and cell intrinsic immunogenicity in a human perspective, which is a nice addition. I do however wonder how easily these data can be generalized. For as far as I can see in various independent datasets the expression levels of HLA-DQA1 in human leukemias and normal stem/progenitor cells is quite diverse, with on average the highest expression in inv16 patients, but in fact in all genetic subgroups there appear to be patients that have either high or low expression of HLA-DQA1. And if anything, in most cases in most datasets there appear to be lower levels of HLA-DQA1 in AML compared to normal HSCs/MPPs.

We thank the reviewer for this opinion and agree that there are limitations on how our findings relating to MHC Class II expression in different genetic subtypes of AML can be generalized with the analysis of diagnostic human samples. We propose that there are 4 major factors that influence the expression of MHC Class II and the other immune-related surface markers we have analysed:

- The stage of differentiation of the AML blast population – MHC Class II expression is greater in HSPCs and appears to decrease with commitment to the myeloid lineage.
- The cell intrinsic effects of the individual oncogene – our analysis of HSPCs subsequent to short-term oncogene expression suggest that Nras^{G12D} is sufficient to drive increased surface expression of MHC Class II and CD86. Recent studies support a hypothesis that this is due to the ability of mutant Ras to drive cell intrinsic IFN signaling (2).
- The presence of a competent immune system – The presence of functional T and NK cells within a cellular microenvironment drives a basal level of interferon signalling that will impact the expression of immunomodulatory surface marker expression.
- The presentation of immunogenic ligands – the presence of an anti-leukemia response will either result in the elimination of cells that are more effective at presenting immunogenic ligands or

select for decreased or increased expression of surface ligands that modulate the anti-leukemic response.

The murine models we have presented allows us to control many of these variables temporally and genetically and provides a model to understand their dynamic interplay that cannot be resolved in a human AML sample at a single time-point in disease evolution.

Q3. 'Regarding the single cell data taken from the van Galen paper: more details could have been provided. Which cells are actually analyzed and plotted in Fig.2s and Suppl. Fig.2C? All cells? Only the AML1-ETO translocated cells?

Thank you for the opportunity to clarify. Due to reviewer comments, the retrospective analysis of the data from van Galen et al (3) has been removed from the modified manuscript. We have now utilised a set of scRNAseq data published in Nature Cancer (4). Here (Figure 2D) we have compared the average expression of HLA-DQA1 in all malignant CD33+ cells in BM samples from AML patients to all CD33+ cells in the healthy BM controls. Malignant cells were defined on the basis of copy number variation and transcriptional profile as described in Lasry et al.(4).

Q4. One wonders why MHC II molecules would be upregulated in leukemic cells? This clearly would not provide any benefit. It has previously also been suggested that cell surface expression of HLA-DR,-DQ and -DP is in fact often lost on leukemia cells, due to downregulation of the HLA class II regulator CIITA. Earlier work also suggested that MHCII is epigenetically repressed in AML cells and that decitabine treatment would impair leukemogenesis by inducing more MHCII. While it is indeed true that in that Dufva et al paper (5) it was suggested that, while in the majority of AML subgroups are HLAII low, AML harboring CBFβ-MYH11 or RUNX1-RUNX1T1 translocations were characterized by higher HLA II expression and CIITA hypomethylation compared to other subtypes, it would be good to explore this in somewhat more detail. It is clear that differences exist between AML subtypes with regard to the level of downregulation of the antigen-presentation response, but whether this is truly increased in AML1-ETO cells compared to normal HSPCs and what the mechanistic link between AML1-ETO, HLAII and possibly CIITA is remains unclear. Line 241: "These data reveal discrete effects of oncogenic drivers on immune regulatory molecule expression in AML cells, supporting a model whereby AE/NrasG12D AML has greater potential to interact with the immune system." Can we truly interpret this as a downstream consequence of oncogenic drivers? I think the authors could attempt to more thoroughly investigate the MHC class II landscape across multiple genetically distinct subgroups of AML patients and link this mechanistically to AML1-ETO.

The reviewer raises a very interesting discussion point. It is now clear from our short-term oncogene expression experiments that the Reviewer suggested, that Nras^{G12D} is sufficient to drive expression of both MHC Class II and the co-stimulatory molecule CD86. MHC expression is upregulated in response to interferon signalling. Recently it has been shown that constitutively active Kras can drive Interferon signalling via de-repression of transposable elements (1). Consistent with this finding, and in combination with our results, microarray analysis of bulk AML (6) shows that mutant Ras AML (green) exhibits a comparatively high level of interferon signalling related transcript expression in comparison to MLL translocated AML (blue).

These data provide support to the conclusion that some oncogenes do indeed drive the expression of high-levels of antigen presentation machinery, presumably as a by-product of the activation of pathways that are required for their oncogenic potential and as such may determine the necessity for cooperation with additional oncogenic lesions for transformation. We do not propose that there is a gain of function associated with increased MHC class II expression.

The cooperative relationship of mutant Ras and elevated expression of Myc in cellular transformation is illustrative of oncogenic cooperation and has been studied in a multitude of cancer types. Of note is that Myc has recently been shown to suppress an interferon-driven transcriptional response in two independent publications on pancreatic ductal adenocarcinoma and triple negative breast cancer (7, 8). The revised manuscript contains additional work clarifying the relationship between Myc and Nras expression. Consistent with these published works, we have now demonstrated that the ectopic expression of Myc in Nras^{G12D}-driven AML is sufficient to drive a decrease in MHC Class I (H2-Db, H2-Kb) and II surface expression (Figure 5E).

Q5. Furthermore, in the last part of the paper the authors go in great detail to study the T cell landscape in human AML1-ETO patients, but whether the described early T cell exhaustion is the consequence of enhanced antigen presentation is not immediately clear.

We apologise for any confusion by the inclusion of these data. In the revised manuscript we have identified a predominant role for Nras^{G12D} driving immune selection and have therefore refocused the paper. Consequently, we have removed the section and related figures in which we characterise the immune microenvironment of human AML1-ETO t(8;21) translocated AML.

Q6. In Figure 2F it is shown that BA/NH had the highest expression of the immunosuppressive ligands PD-L1 and GAL-9 compared to MA9 and AE/NrasG12D, but I guess the authors then conclude that these factors are not dominant since no differences were seen in the transformation process of BA/NH transplanted cells in immunodeficient versus immunocompetent models?

Thank you for raising this important point. The primary AMLs were generated in the absence of a competent immune system by using both Rag2^{-/-}γc^{-/-} donors and recipients (Supplementary Fig. 1B). The comparatively higher levels of PD-L1 and GAL-9 expression observed in the BA/NH AML was determined via the analysis of AMLs that had been passaged through immunodeficient Rag2^{-/-}γc^{-/-} recipients only. We hypothesise that the higher levels of PD-L1 and GAL-9 expression observed comparatively in the BA/NH AML contributes to the inability of a competent immune system to alter BA/NH AML disease progression.

We have referenced these findings in the discussion as follows:

“Furthermore, we found that immune cell ligand expression corresponds to the observed difference in AML immunogenicity, indicating that oncogenes influence the inherent potential of an AML cell to interact with the immune system. Specifically, BA/NH cells had low expression of MHC Class I (H2-D^b and H2-K^b) together with high expression of inhibitory immune checkpoint ligands PD-L1, GAL-9 and CD155 compared to MA9 and Nras^{G12D}. This indicates that the non-immunogenic phenotype of BA/NH may be driven by inherent low antigen presentation and high expression of immune suppressive checkpoint molecules.”

Q7. In the immunoediting experiments described in Fig.3, it is concluded that AML1-ETO+NRasG12D can undergo immunoediting when propagated in immunocompetent mice, but MLL-AF9 cells not. Again, the kinetics of leukemic onset in the MLL-AF9 model are significantly faster, and I wonder whether differences would exist when fewer MLL-AF9 cells were transplanted.

Thank you for the opportunity to clarify these findings. We do not believe that this reflects the input dose of leukaemia cells across the models as AML onset is comparable in the Rag2^{-/-}γc^{-/-} recipients transplanted with either MLL-AF9 or NRas^{G12D} (Figure 4B-C). Conversely, the difference in latency between secondary WT and Rag2^{-/-}γc^{-/-} recipients are dramatic with 2-3 fold prolongation in latency in NRas^{G12D} (Figure 1G). We would also note that when we injected fewer AML cells into recipients in order to extend disease latency in the non-immunogenic BA/NH model, we still failed to observe an increase in disease latency in WT vs Rag2^{-/-}γc^{-/-} recipients (Supplementary Fig. 1H).

Q8. The transcriptional changes that were seen in immunoedited AML1-ETO+NRasG12D cells, were these signatures not seen in MLL-AF9 cells that were propagated in immunocompetent mice versus immunodeficient mice?

We thank the reviewer for this excellent suggestion, which is an extension of Q7 above. We have now compared transcriptional profiles between MA9 AML cells that have been propagated through immunodeficient (Rag2^{-/-}γc^{-/-}) vs. immunocompetent (wild type) recipients and included results in Supplementary Fig. 6G-H and in text. In support of the hypothesis that a reduction in Nras signalling and a compensatory upregulation in Myc-driven transcription is an important immune escape mechanism in IE NRas^{G12D} AML, genes sets related to Myc targets or the down regulation of NRas signalling were not enriched in the MA9 wildtype condition (Supplementary Fig 6G). Interestingly, we observe increased inflammatory signatures in MA9 AML passaged in wildtype vs Rag2^{-/-}γc^{-/-} recipients, consistent with a transcriptional response to the presence of a competent immune system (Supplementary Fig 6H). Interestingly, the same pathways were suppressed in IE (wild type) vs N-IE (Rag2^{-/-}γc^{-/-}) NRas^{G12D} AML (Supplementary Fig 6I).

Q9. Besides retroviral overexpression models the authors also make use of a knockin NPMc/NRas12D model, and this model obviously does not suffer from potential non-physiological high level of oncogene expression. Intriguingly, in this model also no upregulation of H-2Db and MHC Class II molecules was observed, but there an upregulation of ligands with potential immunosuppressive function, PD-L1 and CD86, was observed.

We believe the data the reviewer is referring to is that presented in Figure 4D-E. We would like to clarify that this data is a comparison between the N-IE and IE retroviral Nras^{G12D} model. We have clarified this important point in the revised manuscript.

Q10. What would explain the difference between this knockin model with the AML1-ETO+NRasG12D retroviral model? Does RAS signal differently in the context of AML1-ETO vs NPMc? Does the expression level of RAS matter? It would be of interest to determine whether MHCII expression levels go up upon retroviral transduction with AE, NRASG12D or both, before transplantation in mice.

Thank you again for this suggestion. We have responded to this in Q1 above. To summarise, we have now determined that short-term expression of retroviral Nras^{G12D} is sufficient to drive the increased expression of MHCII in cultures HSPCs prior to transplantation.

Q11. Line 295: "... we observed an inverse relationship between the variant allele frequency of mutant NRAS and the expression of multiple HLA (MHCII) genes, including HLA-DQA1, in NRAS-mutated human AML

[41](Fig. 3H, Supplementary Fig. S3F)". Since MHCII did not go down in IE AE/NRASG12D cells one wonders about the exact molecular pathways downstream of RAS that would drive immunogenicity, or whether more indirect mechanisms would be involved.

Thank you for this question, we believe that the downstream pathways include a number of indirect, generalised responses. As highlighted previously, we have now determined that short-term expression of retroviral Nras^{G12D} is sufficient to drive the increased expression of MHCII in cultured HSPCs prior to transplantation. Recent studies in lung cancer have demonstrated that mutant Ras drives an inflammatory response, thought to be mediated in part by the de-repression of transposable elements (1). However, we must also consider that mutant Ras has been shown to be presented as a tumour neo-antigen and can drive an antigen specific T cell response (9).

We were also surprised that MHCII surface expression did not go down in IE Nras^{G12D} AML. However, when we performed RNAseq analysis on the MA9 AML passaged through either wild type or Rag2^{-/-}γc^{-/-} recipients, we noted that interferon signalling was enriched in the wild type condition, consistent with a response to the presence of a competent immune system (Supplementary Fig 6H). In contrast, interferon signalling was enriched in the N-IE (Rag2^{-/-}γc^{-/-}) condition in Nras^{G12D} AML (Supplementary Fig 6I). We also note that IE Nras^{G12D} AML was enriched for pathways relating to Myc transcriptional activity and the revised manuscript now provides greater focus on the interaction of Myc and Nras in the regulation of immune responses (see response to reviewer 2, points Q1-3).

Q12. line 341 " This indicates that NK cells contribute to immune control of MA9 and AE/NrasG12D AML, whereas T cells play a more important role in immune control of AE/NrasG12D AML." Does the data truly support this notion? Not clear from fig 4A-B, depletion of T cells still enhances tumorigenicity in the MA9 model.

We agree with the reviewer that this comment was not clear. In order to focus the narrative we have now excluded the analysis related to NK-mediated control of AML. We now believe it is easier to appreciate that T cells have a more dominant effect in controlling Nras^{G12D} versus MA9-driven AML.

Q13. Fig 4E-G: How should the reduction in T cells in the BA/NH model be interpreted since there was no difference in disease onset between immunodeficient versus immunocompetent mice?

This is an interesting finding given the lack of an anti-leukemia immune response in the BA/NH model. We note that the immunocompetent BA/NH recipients demonstrated complete effacement of splenic

architecture. We also observed relatively high immunosuppressive ligand expression of the surface of the BA/NH AML, specifically PD-L1, GAL-9 and CD155.

References to these points have been included in the manuscript as follows:

Results section

“We next sought to determine if this differential requirement for T cells in the anti-leukemic response was reflected in the composition of the immune microenvironment. We observed a significant decrease in the frequency of T cells within the microenvironment of the leukemia-bearing spleens of BA/NH recipients compared to MA9 and Nras^{G12D} recipients and naïve controls (Fig. 3E). We note that the BA/NH recipients demonstrated complete effacement of splenic architecture concordant with this loss of normal T-cell populations.”

Discussion section

“We found distinct expression levels of antigen presentation machinery and inhibitory and activating immune cell ligands on the different AMLs in the absence of exposure to a competent immune system. Furthermore, we found that immune cell ligand expression corresponds to the observed difference in AML immunogenicity, indicating that oncogenes influence the inherent potential of an AML cell to interact with the immune system. Specifically, BA/NH cells had low expression of MHC Class I (H2-D^b and H2-K^b) together with high expression of inhibitory immune checkpoint ligands PD-L1, GAL-9 and CD155 compared to MA9 and Nras^{G12D}. This indicates that the non-immunogenic phenotype of BA/NH may be driven by inherent low antigen presentation and high expression of immune suppressive checkpoint molecules.”

Reviewer #2 (Remarks to the Author):

Austin et al provide a comprehensive and thorough description of how specific AML oncogenes either active or suppress the immune response. They demonstrate that specific AML oncogenes control immunogenicity and immune escape through immunoediting. They then show that immunoediting in AML is mediated through transcriptional plasticity, modulation of immunosuppressive molecules and clonal selection. Overall the claims of the paper are supported by convincing data in human and mouse model systems. Several concepts require additional clarification and further support.

Q1. The majority of mouse AML data is generated by the retroviral transduction/transplantation approach, which can result in preferential viral incorporation into distinct progenitor cell states. To what degree are the differential responses to immune escapes mediated by the cell of origin rather than the oncogenic driver?

The reviewer raises the very interesting question as to whether the differentiation status of the AMLs is involved in mediating the anti-leukemia response. Unfortunately, this concept is difficult to test directly as different oncogenic drivers result in the expansion of progenitor cells at different stages of differentiation. In an attempt to overcome this confounding variable, the comparisons of cell surface protein expression has been restricted to CD11b+ AML cells. We have now also looked at the effect of short-term oncogene expression on cell surface protein expression and note that although the results presented are also CD11b oncogene-expressing cells, the phenotypes presented were also observed when analysis was performed on either the bulk or cKit+ restricted oncogene-expressing cell populations.

We have highlighted this specifically in the manuscript:

“Analysis restricted to the AML CD11b+ myeloid population (Supplementary Fig. S2A) showed that AE/Nras^{G12D} was characterized by the highest expression of antigen presentation machinery, H2-D^b, H2-K^b and MHC Class II (Fig. 2A, Supplementary Fig. S2B).”

Q2. The authors suggest that the distinct responses to immune escape in the AML models are in part due to activated MYC. Can suppression of MYC rescue some of the observed phenotypes?

AND

Q3. One of the most interesting observations is that increased MYC signaling and down-regulation of NRAS may contribute to immune escape. The authors should test this concept by suppressing MYC and reactivating the RAS-RAF-MEK pathway in an attempt to restore immune suppression.

We thank the reviewer for raising this interesting point and suggesting this experiment. We have now generated N-IE Nras^{G12D} AML expressing ectopic Myc or the empty vector control. Using this model we have been able to show that increased expression of Myc is sufficient to drive reduced surface expression of MHC Class I (H2-Db, H2-Kb) and Class II, in addition to the increased surface expression of the immunosuppressive ligand PD-L1 (Figure 5E). Myc expression also results in a dramatic reduction in disease latency in immunocompetent wild type recipients that vastly exceeds the difference observed in immunodeficient Rag2^{-/-}γc^{-/-} recipients (Figure 5F). We hypothesize that Myc facilitates immune evasion in Nras^{G12D} AML by inhibiting the transcriptional response to interferon, supported by recent published findings (1, 7, 8). Consistent with this, when we performed retrospective analysis of RNA sequencing in a murine model of MLL-AF9/Nras^{G12D} AML (10), we found that inhibition of Myc expression and transcriptional activity with the Brd4 inhibitor JQ1 resulted in the significant enrichment of pathways related to interferon signalling, included for the reviewer's interest below:

Conditions:

MLL-AF9/NrasG12D AML, 48hrs treatment with 100nM JQ1 (in red)

MLL-AF9/NrasG12D AML, 48hr treatment with DMSO (vehicle control, in blue)

Q4- BA/NH express low levels of MHC class I genes along with high expression of immune check point ligands. What is the suspected mechanism for these specific gene changes?

We would like to note that the BA/NH AMLs, as presented in Figure 2A and E, express comparable levels of MHC Class I (H2-Db, H2-Kb), Class II, CD86 and PD-L1 to wildtype, non-transduced primary CD11b+ cells. They do show induction of GAL-9 and CD155 expression, and to a lesser extent CD80. However, we would note that sustained PD-L1 expression and GAL-9 induction are the only phenotypes unique to this AML. However, we now know that short-term expression of neither BCR-ABL nor NUP98-HOXA9 was sufficient to drive any changes in the expression of these surface markers in culture (Supplementary Fig. 2D and 3B). This would suggest that the immunophenotype of the primary BA/NH AMLs is a function of either co-expression of both BCR-ABL and NUP98-HOXA9 or their growth in vivo.

Q5 - Figure 1(E-G) – Is the disease manifestation in Rag2^{-/-} and C57BL6 the same or different for each of the oncogenic drivers? In other words, is the type of AML similar in the two models?

Thank you for this interesting question. Although these are all acute myeloid leukaemias, there are some subtle differences in disease morphology as defined by haematoxylin and eosin (H&E) staining in the spleens, liver and bone marrow of Rag2^{-/-}γc^{-/-} and WT mice with 2° AMLs. Mice with 2°BA/NH had complete effacement of their splenic and liver architecture. There were no obvious boundaries between the infiltrating leukemic cells and the rest of the spleen, liver or bone marrow. For 2°MA9 mice, splenic

architecture was completely effaced in Rag2^{-/-}γc^{-/-} mice with some tumour borders still observed in WT mice with infiltrating cells located around blood vessels. In the livers of WT and Rag2^{-/-}γc^{-/-} mice, distinct areas of infiltrating MA9 cells can be observed around blood vessels. 2^oNras^{G12D} AML had a distinctive tumour morphology in the spleen and liver with distinct separate lesions and defined tumour borders. Haemorrhagic necrosis in the middle of splenic tumours were observed in Rag2^{-/-}γc^{-/-} mice specifically. Importantly, we observed infiltrating leukocytes in Nras^{G12D} splenic tumours in the WT mice which were not visible in the Rag2^{-/-}γc^{-/-} mice indicating distinct immune interactions in the different immune microenvironments. Uniquely, the bone marrow of Nras^{G12D} Rag2^{-/-}γc^{-/-} mice was infiltrated with AML cells in discrete clusters. In contrast, only a few clusters of AML cells were observed in the bone marrow of the WT mice.

We have included H&E stained sections of AML-bearing recipients below for the reviewers reference:

Reference to the histological presentation of disease has been included in the manuscript as follows:
 “We note that the BA/NH recipients demonstrated complete effacement of splenic architecture concordant with this loss of normal T-cell populations.”

Q6 - Is there an explanation for why MA9 AML is delayed in C57BL6 (relative to Rag2^{-/-}; Figure 1F) but not for BA/NH AML (Figure 1E) even though both AMLs express similar levels of MHC genes (Figure 2)?

We thank the reviewer for raising this point. We would note that the BA/NH AMLs express higher surface levels of the immunosuppressive ligands PD-L1 and GAL-9 in comparison to MA9 which could contribute to an intrinsically immune-suppressive phenotype. Discussion related to this point has been included in the manuscript:

“Specifically, BA/NH cells had low expression of MHC Class I (H2-D^b and H2-K^b) together with high expression of inhibitory immune checkpoint ligands PD-L1, GAL-9 and CD155 compared to MA9 and Nras^{G12D}. This indicates that the non-immunogenic phenotype of BA/NH may be driven by inherent low antigen presentation and high expression of immune suppressive checkpoint molecules.”

Given the high level of MHC Class II expression in our most immunogenic model, we have decided to focus this manuscript on the role of T cells in the anti-leukemia response. However, a number of high-profile studies have detailed the role of NK cells in the control of AML, which are largely considered to function independent of MHC (11).

We have included this point in the discussion:

“It must be noted that MA9 AML exhibited an appreciable immunogenic phenotype despite exhibiting low levels of MHC Class I and II expression, indicating a possible role for NK cells in the control of this leukemia.”

Consistent with this, we have already demonstrated a role for NK cells in the control of MA9 that was removed from the manuscript in order to improve the focus of manuscript as suggested. We have included this below for the reviewer’s interest:

Q7 - The anti-PD1 data (Figure 3M) is modest and highly variable.

We agree with this comment and have highlighted in the revised manuscript that anti-PD1 has limited utility as a single-agent in re-activating the anti-leukemic immune response. These findings are consistent with a number of ongoing clinical trials in AML (12, 13).

The following text has been included in the discussion:

“Although immunoediting was characterized by the upregulation of the immunosuppressive ligand PD-L1, anti-PD1 therapy had limited efficacy in restoring the anti-leukemia immune response in this model, suggesting that the strategy employed in the immunoedited *Nras*^{G12D} AML to evade the immune system is likely to be multifaceted. While ICB clinical trials are ongoing in AML patients, preliminary data suggests that single agent ICB also has minimal activity in AML patients whereas ICB in combination with hypomethylating agent azacitidine has shown positive responses in a proportion of AML patients with higher pretherapy bone marrow CD3+ and the presence of ASXL1 mutation, again suggesting that patient genetic profiles are important predeterminants of treatment efficacy (12, 13)”

Reviewer #3 (Remarks to the Author):

Austin et al provide a comprehensive and high-quality manuscript. To my knowledge*, the data are technically sound, appropriately analysed and interpreted. Appropriate controls have been used. Their conclusions are sound and claims not overstated. Contextualization in introduction and discussion is excellent and accessible to a broad audience. The manuscript is very clearly written with enough detail to understand the experiments in nearly all places. They provide novel data on the relationship between oncogenic driver and immune control of AML and detailed studies to understand the mechanisms of immunogenicity and immune escape. As the authors point out, immune-directed therapies potentially do have a role in this disease, if used in combination and in the right circumstances. Therefore, this paper contains information important to advancing personalized therapy for AML

*I do not have expertise in performing mouse experiments, so cannot comment on methodology of model generation/transfer and/or strain selection.

Key results are that immunocompetent mice survived longer in MA9 but especially in AE/NRas disease. This is accompanied by MHC class II expression higher on AE/NRas than MA9 in models, AML patients at

bulk tumour and single cell level. Furthermore, AE/NRas AML generated on an MHC II background had shorter survival than when generated on a WT background. They authors found accelerated disease in immunocompetent mice with AE/NRas having previously passed through an immunocompetent (vs an immunodeficient host), accompanied by increased MYC expression, reduced NRas expression and reduced mutant NRas copy number. They established that NK/T depletion accelerated disease progression in AE/NRas and that AE/NRas in co-culture stimulated T cell proliferation. Differences in tumour environment T cells were not particularly marked but increase in CD8 Effector proportion in AE/Ras was noteworthy. Alterations in immunosuppressive molecule expression was found in AE/NRas CD8 T cells.

scRNAseq data on immune-enriched AML1-ETO BM added detail and supported these observations.

Specific points

Q1. Please make it a little clearer at the start of the results which mutations represent which risk category. Is there any reason these models of AML were chosen besides fitting into favourable, intermediate and adverse prognostic subclasses? Specifically, NUP98-HOXA9 is a rare translocation and I was curious why this was selected.

Thank you for the opportunity to address this important point. The oncogenes were initially selected as they are well established models of AML and have been characterised extensively in the preceding literature. Moreover, these models were able to engraft non-irradiated recipients in our hands, whereas other models (including CDX2 overexpression) did not engraft in non-irradiated recipients. The BCR-ABL/NUP98-HOXA9 is admittedly a rare subtype of AML, however to our knowledge is one of the few models of adverse prognostic disease that is able to be modelled using murine systems. We have performed preliminary experiments with FLT3 expressing AML samples and have observed that these are not controlled by immune control as potently as Ras expressing AML samples, however these experiments remain preliminary and are part of a separate project / collaboration. Moreover, we have recently identified important off target effects of the FLT3 expressing AML model (under second revision at Leukemia journal) which raise a number of concerns about the interpretation of these data for such experiments. Thus, the AML oncogenes were selected on the basis that these are representative and tractable models of AML. We are careful not to extrapolate these findings to all patients with adverse (or favourable disease) and our main conclusions in the revised manuscript are based around the functional interplay between the RAS and MYC pathways.

Q2. Why do you think it is that primary MA9 and AE/NRas mice survive 40-80 days post-transplant (Fig S1B) but secondary mice also on a Rag2^{-/-}γc^{-/-} background survive 20 days (Fig 1F)?

One of the major contributing factors to the differences in latency observed in primary and secondary recipients is that primary mice were injected with fewer oncogene-expressing (GFP+) cells compared to secondary recipients. We should also consider that the cells transplanted into primary recipients are freshly transduced primary HSPCs. Although this oncogenic combination is appreciated as sufficient to drive malignant transformation, it is thought that this capacity is restricted to a specific stem or progenitor stage (14). In contrast, secondary transplants utilise a population of transformed cells with a proven replicative capacity and that have been enriched for leukaemia stem cell activity. Therefore, the shorter latency in the secondary recipients presumably reflects enrichment of leukaemia stem cell activity and a fully transformed progenitor population. This finding has been described in the literature previously (14).

Q3. Sup Figure 2. In understand from Fig S1 that 11b expression should capture most of the myeloblasts. However the % of 11b+ cells in BA/NH in Fig S2A seems very low. To understand this it would be helpful to clarify in the legend what tissue is being sampled and at what timepoint following transfer? Are you sure the CD11b gate is comprehensively sampling BA/NH blasts? Sup Figure 2D How many times performed?

Thank you for the opportunity to clarify, all surface marker analysis was performed on splenocytes isolated from Rag2^{-/-}γc^{-/-} recipients at a near terminal disease stage. The analysis was performed once but included the comparison of four independently generated BA/NH AMLs.

The modified Figure 2 legend is as follows:

“(A) Median fluorescence intensity (MFI) of H2-D^b, H2-K^b, MHC Class II (IA/E) on cell surface of myeloid cells (CD11b⁺) from bone marrow (BM) of naïve wildtype mice and GFP⁺ CD11b⁺ BA/NH, MA9 and Nras^{G12D} AML cells in the spleens of Rag2^{-/-}γc^{-/-} recipients moribund with disease.”

“(E) MFI of CD80, CD86, PD-L1, GAL-9 and CD155 on cell surface of myeloid cells (CD11b⁺) from BM of naïve wildtype mice and GFP⁺ CD11b⁺ BA/NH, MA9 and Nras^{G12D} AML cells in the spleens of Rag2^{-/-}γc^{-/-} recipients moribund with disease.”

“Each point represents a biological replicate derived from independent mice transplanted with the same tumor.”

We agree with the reviewer that the CD11b⁺ gate in Supplementary Fig. 2A does not comprehensively sample the BA/NH blasts. Analysis was performed on this lineage restricted population to control for the potential effects of differentiation status on surface marker expression. However, we would note that differences in surface marker expression observed are largely consistent when we instead perform the analysis on the bulk oncogene-expressing (GFP⁺) population for the BA/NH AML only, included below for your reference. Of note, is a relative increase in the expression of MHC Class I and a significant increase in PD-L1 expression when compared to WT.

Populations compared are as follows:

WT BM (CD11b⁺), BA/NH (all GFP⁺), MA9 and NrasG12D (GFP⁺/CD11b⁺)

Q4. I cannot locate the figure panel providing evidence that there is no intrinsic difference in proliferative capacity associated with IE. We are referred to Fig 3B but that is a survival curve.

We appreciate the reviewers comment and the need for clarification. We had originally proposed that similar survival in Rag2^{-/-}γc^{-/-} recipients between IE and N-IE leukaemias was indicative of a similar proliferative capacity in vivo (Figure 4B). To further support this conclusion, we have now directly examined cell proliferation using IHC for the proliferative marker Ki67 and the mitotic marker phosphorylated histone H3 (p-H3). Nras^{G12D} AML in Rag2^{-/-}γc^{-/-} and wildtype recipients at a near-terminal disease stage show a small increase in Ki67 in the tumour regions of the spleen of wild type

recipients, but no significant difference for either marker in tumour regions of the liver or p-H3 in tumour regions of the spleen. These data have been included in the modified manuscript (Supplementary Fig. 6E-F). In aggregate, these results support the statement that there is no significant difference in the proliferative capacity of IE and N-IE AML.

Q5. The authors use a creative way to look at correlation between MYC overexpression and immune environment, but it requires more explanation and the correlation is not wholly convincing. It is clear where the score comes from (ref 39). Where is the AML1-ETO data from? What is it- BM?. How is Myc measured here?

We agree with the reviewer that the explanation of this analysis could be clearer. We note that the revised figure is now Figure 5D in which we have used a mutant NRAS AML data set GSE6891 (6). The microarray analysis in this dataset was performed on blasts purified from both bone marrow and peripheral blood of patients with AML. The Myc core gene set is published (15) and represents a discrete set of Myc targets that are upregulated in response to the acute activation of Myc across a broad panel of adult tissues. An enrichment score for this gene set is determined for each individual AML sample using single sample gene set enrichment analysis (16). We have now clarified this in the text. We also note that in the revised manuscript, this data is not intended to stand alone but to serve as a companion data with which the human relevance of the functional murine data presented can be addressed.

For ease of reference, the modified figure legend is as follows:

“(D) Correlation between the relative enrichment of a gene set containing core Myc transcriptional targets (15) and genes associated with cytolytic infiltrate in AML (5), using bulk expression data from NRAS mutant AML patients (6).”

The data is referenced in the results section of the manuscript as follows:

“Consistent with this, in NRAS-mutant AML, we observed that a tissue-agnostic set of Myc transcriptional targets showed an inverse correlation with a predictive gene signature of cytotoxic immune cell infiltration in AML (Fig. 5D) (5, 15)”

Q6. A visual representation of the overall findings would be a worthwhile addition

Thank you. We agree and are happy to provide a visual abstract to accompany the article if accepted.

1. Freedman AS, Freeman GJ, Rhyhart K, Nadler LM. Selective induction of B7/BB-1 on interferon-gamma stimulated monocytes: a potential mechanism for amplification of T cell activation through the CD28 pathway. *Cell Immunol.* 1991;137(2):429-37.
2. Reggiardo RE, Maroli SV, Halasz H, Ozen M, Hrabeta-Robinson E, Behera A, et al. Mutant KRAS regulates transposable element RNA and innate immunity via KRAB zinc-finger genes. *Cell Rep.* 2022;40(3):111104.
3. van Galen P, Hovestadt V, Wadsworth li MH, Hughes TK, Griffin GK, Battaglia S, et al. Single-Cell RNA-Seq Reveals AML Hierarchies Relevant to Disease Progression and Immunity. *Cell.* 2019;176(6):1265-81 e24.
4. Lasry A, Nadorp B, Fornerod M, Nicolet D, Wu H, Walker CJ, et al. An inflammatory state remodels the immune microenvironment and improves risk stratification in acute myeloid leukemia. *Nat Cancer.* 2022.
5. Dufva O, Polonen P, Bruck O, Keranen MAI, Klievink J, Mehtonen J, et al. Immunogenomic Landscape of Hematological Malignancies. *Cancer cell.* 2020;38(3):380-99 e13.

6. Verhaak RG, Wouters BJ, Erpelinck CA, Abbas S, Beverloo HB, Lugthart S, et al. Prediction of molecular subtypes in acute myeloid leukemia based on gene expression profiling. *Haematologica*. 2009;94(1):131-4.
7. Muthalagu N, Monteverde T, Raffo-Iraolagoitia X, Wiesheu R, Whyte D, Hedley A, et al. Repression of the Type I Interferon Pathway Underlies MYC- and KRAS-Dependent Evasion of NK and B Cells in Pancreatic Ductal Adenocarcinoma. *Cancer Discov*. 2020;10(6):872-87.
8. Zimmerli D, Brambillasca CS, Talens F, Bhin J, Linstra R, Romanens L, et al. MYC promotes immune-suppression in triple-negative breast cancer via inhibition of interferon signaling. *Nat Commun*. 2022;13(1):6579.
9. Tran E, Robbins PF, Lu Y-C, Prickett TD, Gartner JJ, Jia L, et al. T-Cell Transfer Therapy Targeting Mutant KRAS in Cancer. *The New England journal of medicine*. 2016;375(23):2255-62.
10. Zuber J, Shi J, Wang E, Rappaport AR, Herrmann H, Sison EA, et al. RNAi screen identifies Brd4 as a therapeutic target in acute myeloid leukaemia. *Nature*. 2011;478(7370):524-8.
11. Paczulla AM, Rothfelder K, Raffel S, Konantz M, Steinbacher J, Wang H, et al. Absence of NKG2D ligands defines leukaemia stem cells and mediates their immune evasion. *Nature*. 2019;572(7768):254-9.
12. Liao D, Wang M, Liao Y, Li J, Niu T. A Review of Efficacy and Safety of Checkpoint Inhibitor for the Treatment of Acute Myeloid Leukemia. *Front Pharmacol*. 2019;10:609-.
13. Daver N, Garcia-Manero G, Basu S, Boddu PC, Alfayez M, Cortes JE, et al. Efficacy, Safety, and Biomarkers of Response to Azacitidine and Nivolumab in Relapsed/Refractory Acute Myeloid Leukemia: A Nonrandomized, Open-Label, Phase II Study. *Cancer Discov*. 2019;9(3):370-83.
14. Krivtsov AV, Twomey D, Feng Z, Stubbs MC, Wang Y, Faber J, et al. Transformation from committed progenitor to leukaemia stem cell initiated by MLL-AF9. *Nature*. 2006;442(7104):818-22.
15. Bywater MJ, Burkhart DL, Straube J, Sabo A, Pendino V, Hudson JE, et al. Reactivation of Myc transcription in the mouse heart unlocks its proliferative capacity. *Nat Commun*. 2020;11(1):1827.
16. Hanzelmann S, Castelo R, Guinney J. GSEA: gene set variation analysis for microarray and RNA-seq data. *BMC Bioinformatics*. 2013;14:7.

REVIEWERS' COMMENTS

Reviewer #1 (Remarks to the Author):

The efforts of the authors to address my comments are much appreciated

Reviewer #2 (Remarks to the Author):

The authors have addressed my salient concerns. The revised version of the manuscript is significantly improved. Congratulations to the authors on an excited finding.

Reviewer #3 (Remarks to the Author):

I am happy with the responses authors have made to my original comments and the clarifications they have made to the manuscript. Significant additional work has been undertaken, allowing the authors to make more specific conclusions the role of NrasG12D and about Myc in immune responses to AML. The resulting manuscript is strong and with important implications for AML therapeutics. I have no further comments to make.

RESPONSE TO REVIEWERS' COMMENTS

Reviewer #1 (Remarks to the Author):

The efforts of the authors to address my comments are much appreciated. We thank the reviewer for their time. We greatly appreciate their insights and valuable suggestions that contributed greatly to the improvement of this manuscript.

Reviewer #2 (Remarks to the Author):

The authors have addressed my salient concerns. The revised version of the manuscript is significantly improved. Congratulations to the authors on an excited finding. We thank the reviewer for their time. We greatly appreciate their insights and valuable suggestions that contributed greatly to the improvement of this manuscript.

Reviewer #3 (Remarks to the Author):

I am happy with the responses authors have made to my original comments and the clarifications they have made to the manuscript. Significant additional work has been undertaken, allowing the authors to make more specific conclusions the role of NrasG12D and about Myc in immune responses to AML. The resulting manuscript is strong and with important implications for AML therapeutics. I have no further comments to make. We thank the reviewer for their time. We greatly appreciate their insights and valuable suggestions that contributed greatly to the improvement of this manuscript.